# An Overview of the Potential of Food-Based Carbon Dots for Biomedical Applications

**DOI:** 10.3390/ijms242316579

**Published:** 2023-11-21

**Authors:** Chen-Yow Wang, Nodali Ndraha, Ren-Siang Wu, Hsin-Yun Liu, Sin-Wei Lin, Kuang-Min Yang, Hung-Yun Lin

**Affiliations:** 1Department of Bioscience and Biotechnology, National Taiwan Ocean University, Keelung 202301, Taiwan; 101127xd@gmail.com (C.-Y.W.); nodali@email.ntou.edu.tw (N.N.); tiggermss818@gmail.com (H.-Y.L.); andy920140@gmail.com (S.-W.L.); ab0913669898@gmail.com (K.-M.Y.); 2Department of Food Science, National Taiwan Ocean University, Keelung 202301, Taiwan; 3Division of Microbiology and Immunology, Graduate Institute of Biomedical Sciences, College of Medicine, Chang Gung University, Taoyuan 333323, Taiwan; jerry0127people2017@gmail.com; 4Center of Excellence for the Oceans, National Taiwan Ocean University, Keelung 202301, Taiwan

**Keywords:** food-based carbon dots, therapeutic activities, human healthcare, biosafety assessment, carbon dots family

## Abstract

Food-based carbon dots (CDs) hold significant importance across various fields, ranging from biomedical applications to environmental and food industries. These CDs offer unique advantages over traditional carbon nanomaterials, including affordability, biodegradability, ease of operation, and multiple bioactivities. This work aims to provide a comprehensive overview of recent developments in food-based CDs, focusing on their characteristics, properties, therapeutic applications in biomedicine, and safety assessment methods. The review highlights the potential of food-based CDs in biomedical applications, including antibacterial, antifungal, antivirus, anticancer, and anti-immune hyperactivity. Furthermore, current strategies employed for evaluating the safety of food-based CDs have also been reported. In conclusion, this review offers valuable insights into their potential across diverse sectors and underscores the significance of safety assessment measures to facilitate their continued advancement and application.

## 1. Introduction

The synthesis of carbon nanotubes in 1985 and the subsequent isolation of graphene and carbon dots (CDs) in 2004 have been significant milestones in the advancement of carbon nanomaterials (CNMs) [1,2]. These materials possess exceptional properties that have revolutionized diverse industries, fostering scientific exploration and technological progress. For nearly 20 years, extensive research has been conducted on the physicochemical characteristics, synthesis methodologies, and wide-ranging applications of CNMs, placing them at the forefront of nanotechnology, energy, and materials science research [3]. Among them, the potential of CDs in biomedicine is gaining more and more interest due to their remarkable biocompatibility, light-emitting characteristics, drug delivery capabilities, immunomodulatory effects, and antimicrobial activity [3,4,5,6]. These characteristics position CDs as valuable tools in various clinical domains, including bioimaging, targeted drug delivery, pathogen control, and immunotherapy.

Typical CDs are regarded as organic carbonization products with sizes less than 20 nm, exhibiting excitation-dependent fluorescence properties [7]. They possess sp^2^/sp^3^ carbon skeletons and feature an abundance of functional groups and polymer chains within their structures [8]. The surface of CDs is rich in hydrophilic compounds, including carboxyl, hydroxyl, and amine groups, contributing to their excellent water dispersibility [9]. Before 2019, CDs could be further categorized based on their structure (Figure 1), including graphene quantum dots (GQDs), carbon quantum dots (CQDs), carbon nanodots (CNDs), and carbonized polymer dots (CPDs) [10]. Among these, GQDs are characterized as two-dimensional materials with layered structures typically less than 20 nm in width, generally extending up to five layers (ca. 2.5 nm) [11]. Their primary planar structure consists of sp^2^ carbon hybrid arrangements, predominantly at the edges of graphene sheets or within interlayer defects [12]. Notably, GQDs exhibit distinct graphene lattice structures and a significant presence of chemical groups, particularly oxygen-containing functional groups, contributing to their unique properties, such as the quantum confinement effect and edge effect [13]. CQDs typically assume a spherical shape, with their carbon core primarily featuring excellent sp^2^ carbon crystallinity and sizes typically falling within the range of 1 to 10 nm [14]. The structural properties of CQDs enable them to exhibit intrinsic state luminescence and size-dependent quantum confinement effect. CNDs closely resemble CQDs in terms of size and shape [15]. They exhibit a higher degree of carbonization but typically lack a distinct lattice structure and do not manifest the quantum confinement effect related to particle size. The photoluminescence of CNDs stems from defects/surface states and subdomain states within the graphitic carbon core [16]. CPDs are produced through the carbonization of polymer compounds, with a relatively low degree of carbonization in their carbon core [10]. Their primary shared characteristic is the surface of the carbon core being enriched with outward-extending polymer functional groups, a result of passivation during the carbonization process. The photoluminescence of CPDs mainly originates from surface states, subdomain states, molecular states, and the crosslink-enhanced emission (CEE) effect [10].

Recently, researchers have successfully synthesized and published several new classes of CDs, including carbonized nanogels (CNGs), carbon-dot liposomes (CD_somes_), and carbon nanovesicles (CNVs), which share many characteristics in line with CDs but also exhibit distinct structural differences (Figure 1). CNGs, with sizes ranging from about 100 to 500 nm, are slightly larger than the typically defined in CDs [17,18]. They feature a carbonized structure comprising sp^2^-conjugated aromatic rings and sp^3^ polymer structures. Due to their graphene-like embedded polymer structure, CNGs can adopt either spherical or irregular-edged particle forms, displaying physical properties characterized by rheological properties similar to those of flexible polymer structures [17]. The photoluminescent characteristics of CNGs primarily stem from the π-conjugated macrocycle structure and edge chemical functional groups. Specifically, the reduction in vibration and rotation of subfluorophores within crosslinked gel structures is thought to trigger the CEE effect [17,18]. In addition, CD_somes_ are the carbonization products of long-chain hydrophobic compounds, whose carbon core structure comprises a conjugated benzene ring formed by a blend of sp^2^/sp^3^ carbon structure with oxygen-containing functional groups on the surface [19]. A significant characteristic of CD_somes_ is the asymmetric retention of the aliphatic chain from the precursor on their surface, giving rise to both hydrophilic and hydrophobic properties, rendering them amphipathic CDs with an approximate size of 5 nm. An even more significant and readily observable characteristic is the self-assembly of CD_somes_ in aqueous solutions, forming vesicles encased by a unilamellar bilayer of amphiphilic CDs. This structure bears a resemblance to liposomes and typically measures around 100 nm in size. The as-formed structure in water is attributed to amphiphilic interactions among the surface ligands, particularly including hydrophobic interactions between the oleate groups [19]. Based on their vesicle structure, CD_somes_ demonstrate excellent photostability, fusogenicity, and biocompatibility in aqueous solutions. Their excitation-dependent fluorescence properties can be attributed to the presence of polycyclic aromatic clusters, surface emissive traps, and edge defects present in amphipathic CDs of various sizes encapsulated within the vesicle structure [20]. CNVs, as the carbonization products of nonionic surfactants, also possess amphipathic CD characteristics with sizes typically around 3–5 nm [21,22]. Their vesicle structures, however, differ from CD_somes_ that self-assemble in aqueous solutions, as CNVs feature multilayered bilayer amphipathic CDs and exhibit structural characteristics resembling lipid nanoparticles. Nevertheless, the limited and potent evidence available does not allow for a clear differentiation in the mechanistic distinctions of photoluminescence characteristics. Hence, the classification of CD_somes_ and CNVs remains controversial.

The synthesis of CDs can be broadly classified into two main approaches: top-down and bottom-up strategies. The top-down approach involves the use of larger carbon substrates, such as graphite powers, graphite sheets, or carbon nanotubes (CNTs), which are prepared through methods like arc discharge, laser ablation, or electrochemical oxidation [23,24]. However, these methods are intricate and energy-intensive. In contrast, the bottom-up approach utilizes small molecular compounds, natural products, and even plant or food sources such as amino acids, organic acids, sugars, flavonoids, edible plants, or fruit juices [23]. By subjecting these materials to external heat, they undergo dehydration, condensation, and catalytic reactions, resulting in carbonization and the formation of CDs. The heat can be supplied through hydrothermal treatment (uniform heating in a solvent), microwave treatment (employing microwaves to facilitate reactions), or combustion (direct calcination) [3,23]. These strategies are known for their operational simplicity, scalability for mass production, and utilization of relatively low-cost instrumentation [3,4,5,6].

Interestingly, the heating conditions employed in these bottom-up strategies bear a resemblance to certain food processing techniques, such as stewing (prolonged closed heating at 60–120 °C for 1–5 h), frying/roasting (brief open heating at 100–300 °C for 5–60 min), and microwave cooking (600–1200 W for 1–10 min) [23,25]. Similarities in substrate abundance and synthetic strategies have led to investigations into the extraction of CDs from various processed foods using organic extraction methods [26,27,28,29,30,31,32,33,34,35,36]. Several studies have observed that substances like beer, instant coffee powder, roast duck, and even Traditional Chinese medicine derived from natural medicinal plants through decoction, baking, or roasting, have the potential to yield CDs [30,31,37,38,39,40,41]. In this review, CDs obtained using food-grade raw materials through food-like processing methods are regarded as food-based CDs. To date, a wide range of food-based CDs has been developed and extensively evaluated as highly biocompatible biomedical materials using diverse animal models.

In the last decade, the application potential of CDs in various fields has been studied and reviewed [3,4,5,6]. However, these studies have little attention to food-based CDs. To respond to this gap, this article provides a comprehensive overview of the recent developments in food-based CDs (Figure 2). This review mainly focuses on the characteristics and properties of food-based CDs. The therapeutic applications of food-based CDs in the field of biomedicine, along with methods for assessing their safety, are also reviewed.

## 2. Food-Based CDs

### 2.1. CDs from Processed Food and Beverages

Thermal treatment is one of the common methods used in food processing to reduce microbial activity and control the presence of foodborne pathogens [42]. During thermal processing, carbonization reactions are often observed in foods, which could result in the formation of CDs [43,44]. Precursors typically undergo a series of carbonization steps, and certain studies offer time-dependent structural analyses illustrating this evolution (Figure 3). Initially, the precursor undergoes dehydration, leading to the aggregation and mild condensation of decomposition products, resulting in the formation of large-sized polymer supramolecular structures [44]. With the progression of heating, these polymer supramolecular structures contract due to ongoing intramolecular dehydration. This process is accompanied by the formation of carbon–carbon bonds and the development of aromatic clusters within the polymer. Once the density of clusters reaches a critical supersaturation point, the nucleation of the carbon core occurs. At this stage, aromatic clusters diffuse toward the particle surface to form nuclei, and the passivation of various functional groups occurs simultaneously [45]. To synthesize CPDs, CNGs, CD_somes_, and CNVs, besides selecting suitable precursors, precise control of temperature and heating duration at this stage is crucial [10,17,18,19,20,21,22]. This control is essential for preserving effective functional groups and stabilizing the carbonized structure. In the case of other types of CDs exhibiting obvious/classical crystal lattices, polymer nanoparticles tend to dissipate or undergo transformation with increasing heating time [44]. This leads to a decrease in the polymer-to-dots ratio, resulting in smaller particle sizes and a narrower distribution of CDs (Figure 3).

Numerous studies have provided insights into the extraction of CDs from thermally processed foods like stewed, roasted, or grilled lamb chops, beef, duck, chicken, eel, salmon, or hook snout carp [26,27,28,29,30,31,32,33,34,35,36]. Typically, the roasting process involves heating raw food in an oven at temperatures ranging from 100–350 °C for 5–60 min [46]. Grilling is a similar process that involves direct heating of the food on a metal grill. Stewing, on the other hand, entails slowly and gently heating raw food in a covered pot with a generous amount of broth at temperatures ranging from 60–80 °C for a duration of 3–5 h, or at 110–120 °C for 1–2 h [25]. For example, Geng et al. stewed beef in a pressure cooker at 117 °C and observed the yields of CDs as high as 0.05, 0.06, and 0.07% (*v*/*v*) after 30, 50, and 70 min of stewing, respectively [29]. The frying process, which involves submerging the raw food in vegetable or animal cooking oil, requires heat at temperatures around 150–200 °C for 10–15 min [25,47]. These heating conditions aligned with the synthesis requirements for most types of CDs in terms of temperature [23]. However, the allotted time is often insufficient to achieve complete carbonization of the raw materials, resulting in low yields of CDs [23,48].

The extracted CDs from processed foods are typically obtained using organic solvents, such as methanol or ethanol, and possess fluorescent properties [26,27,28,29,30,31,32,33,34,35,36,48,49,50,51,52,53,54,55,56,57,58]. Size-based separation techniques, such as dialysis and column chromatography, showed that food-based CDs exhibit sizes ranging from approximately 0.9 to 54.8 nm (Table 1). Interestingly, more complex processed foods, including pizza, burger meat, and canned foods, have also been reported to contain CDs with sizes ranging from 1.8 to 5.8 nm [49,50,51]. Other studies reported that commercially available beverages, including Nescafé, Coke, and Pepsi, also exhibit the presence of CDs, with particle sizes ranging from 0.9 to 39.1 nm [39,52,53,54]. These CDs in the beverages may originate from flavor enhancers that undergo high-temperature processing. Notably, caramel, one of the main ingredients in many commercial beverages, could also produce CDs. Studies showed that CDs in commercial beverages (2–5% *w*/*v*) were slightly higher than those found in other processed foods (typically ranging from 0.3% to 1.0% (*w*/*v*)) [26,27,28,29,30,31,32,33,34,35,36,48,49,50,51,52,53,54,55,56,57,58,59].

Furthermore, food-based CDs have also been identified in various fermented food products, including beer, bread, vinegar, soybean sauce, and tofu wastewater [37,38,48,55,56,57,58,59]. These CDs are believed to result from enzyme conversion reactions facilitated by probiotic microbes. The content of CDs in these foods typically ranges from 0.01% to 1.5%, similar to that found in heat-processed foods (Table 1). Interestingly, honey, a natural polysaccharide produced through the fermentation of plant nectars by microorganisms and enzymes in bees’ mouths, has also been found to contain CDs through dialysis and acetonitrile precipitation [60].

The in vivo synthesis of CDs through artificial induction strategies remains an unresolved difficulty. CD synthesis entails the decomposition, dehydration, and polymerization of organic molecules or polymers (Figure 3). Additionally, the verification of the hypothesis regarding CD induction in microorganisms is particularly challenging due to the constraints of in vitro assays in accurately simulating the intricate interplay of multienzymatic dehydration and carbon bond polymerization reactions involved [43,44]. As of now, no studies have discovered CDs produced by living organisms themselves. Notably, through a phytosynthesis process at 50 °C using plant leaf extracts in an in vitro environment with chitosan dissolved in an acidic solution, chitosan particles (ca. 10 nm) are synthesized [61,62]. While chitosan particles lack a crystal lattice and thus cannot be classified as CDs, they still illustrate the potential of bioenzyme catalysis in the production of novel CDs. The reaction is thought to potentially encompass the condensation and polymerization of several enzymes, including nitrate reductase, β-glucosidase, glycolytic enzymes, and aldolases [63]. Consequently, confirming this hypothesis continues to be a significant undertaking in the field of CD synthesis.

**Table 1 ijms-24-16579-t001:** Examples of CDs extracted from processed food and beverages.

Food Groups	Food Source	Purification	Type/Size (nm)	Yield (%)	Quantum Yield (%)	Toxic Evaluation	Ref.
Complex processed foods	Pizza	Ethanol for 12 h, then dialyzed (0.5 kDa)	CNPs/2.6–4.1	NA	2.1	**In vitro** >1 mg/mL, 6 h (Caco-2 cells) **In vivo** >100 mg/mL, 48 h (*C. elegans*)	[49]
	Burger meat (beef)	Ethanol for 12 h, then dialyzed (3.5 kDa)	CDs/0.9–54.8	NA	23.3	**In vitro** >3.2 mg/mL, 12 h (MO cells) **In vivo** >3.2 mg/mL, 12 h (bean)	[50]
	Canned yellow croaker	Ethanol for 12 h, then dialyzed (1 kDa)	CDs/1.8–5.8	0.3 (*w*/*w*)	9.7	**In vitro** >0.25 mg/mL, 12 h (HepG2 cells)	[51]
Commercial beverages	Nescafé^®^ coffee	Size exclusion (Sephadex G-25)	CDs/3.0–6.0	2.0 (*w*/*w*)	5.5	**In vitro** >20 mg/mL, 24 h (CHO cells)/>1.5 mg/mL, 24 h (SMMC-7721 cells) **In vivo** >1000 mg/g, 56 h (guppy fish)	[39]
	ILLY^®^ coffee	Dialyzed (14 kDa)	CQDs/2.0–7.0	NA	NA	NA	[52]
	Cola	Size exclusion (Sephadex G-25)	CNPs/3.9–5.5	3.0 (*w*/*v*)	NA	**In vitro** >20.0 mg/mL, 24 h (CHO cells) **In vivo** >2000 mg/g, 24 h (mice)	[53]
	Beverages	Size exclusion (Sephadex G-25)	CDs/2.8–39.1	2.0–5.0 (*w*/*w*)	1.5–11.9	**In vitro** >20 mg/mL, 24 h (CHO cells)/>10 mg/mL, 24 h (Tca-8113 cells) **In vivo** >40 mg/mL, 6 h (onion)	[54]
Fermented food products	Beer	Size exclusion (macroporous resin)	CDs/0.9–4.1	NA	1.4–3.9	**In vitro** >5 mg/mL, 4 h (MC3T3-E1 cells) **In vivo** >2000 mg/kg, 24 h (mice)	[37]
	Tsingtao^®^ beer	Size exclusion (Sephadex G-25)	CDs/1.0–5.0	1.2 (*w*/*v*)	7.4	**In vitro** >50 mg/mL, 48 h (MCF-7 cells)	[38]
	Bread	Methanol for 10 min, then dialyzed (1 kDa)	CNPs/21.4–33.6	NA	1.2	**In vitro** >2 µg/mL, 24 h (HeLa cells)	[55]
	Bread	Methanol for 1 h, then dialyzed (1 kDa)	CNs/5.0–20.0	NA	NA	**In vitro** >400 µg/mL, 48 h (hMSCs cells)	[56]
	Breadcrumbs	Ethanol for 12 h, then dialyzed (3.5 kDa)	CDs/2.2–3.2	0.013 (*w*/*v*)	1.8	NA	[48]
	Vinegar	Size exclusion (macroporous resin)	CNPs/1.2–6.2	1.5 (*w*/*v*)	5.7	NA	[57]
	Vinegar	Ethanol for 12 h, then dialyzed (1 kDa)	CNPs/142.6–281.2	NA	NA	**In vitro** 100 µg/mL, 24 h (Caco-2 cells)	[58]
	Soybean sauce	Ethanol for 12 h, then dialyzed (1 kDa)	CNPs/298.5–398.2	NA	NA	**In vitro** 100 µg/mL, 24 h (Caco-2 cells)	[58]
	Tofu wastewater	Ultrasonic shock for 5 min, then centrifuged	CDs/2.0–10.0	NA	NA	NA	[59]
Flavor enhancers	Caramels	Methanol for 10 min, then dialyzed (1 kDa)	CNPs/2.8–5.8	NA	0.6	NA	[55]
	Jaggery	Methanol for 10 min, then dialyzed (1 kDa)	CNPs/12.8–27.8	NA	0.6	**In vitro** >2 µg/mL, 24 h (HeLa cells)	[55]
	Honey	Dialyzed for 48 h, treatment with acetonitrile, then lyophilized	CDs/1.7–4.7	1.5 (*w*/*w*)	1.6	NA	[60]

NA—not available; CDs—carbon dots; CNPs—carbon nanoparticles; CNs—carbon nanostructures; CNTs—carbon nanotubes; CQDs—carbon quantum dots.

### 2.2. CDs Synthesized from Raw Food or Edible Plants

Raw foods, such as meat, vegetables, and fruits, are easily accessible sources of carbon compared to nature-derived chemicals. The bottom-up approach, encompassing dry burning, hydrothermal methods, or microwaving, is commonly employed to carbonize food and thereby generate CDs [23]. To obtain CDs from processed edible plants using the dry burning method, these materials are typically dried, and the resulting homogenized powders are heated in a muffle furnace at temperatures ranging from 180 °C to 400 °C for 1 to 6 h [64,65,66]. The original material undergoes proper oxidation during this heating process, facilitating the dehydration and polymerization of various carbon-containing functional groups [23,67]. Applying appropriate heating facilitates the thermal activation of reactions, supplying the necessary energy for chemical interactions between reactant molecules. This promotes the formation of sp^2^ hybridization and the development of a hexagonal carbon framework [68]. Furthermore, the heating-induced oxidative processes, driven by the electronegativity of oxygen atoms, lead to chemical bonding between carbon atoms, resulting in the introduction of various functional groups and heightened chemical reactivity [69]. However, prolonged heating during dry burning can lead to excessive carbonization, causing the dissipation of non-carbon atoms such as N, O, P, and S [70]. Hence, precise time control during the dry burning process is a critical factor in achieving a rich array of active functional groups on the surface of CDs.

The dry-burning method, also known as powder carbonization, has received limited research attention, and the verification assays related to the graphite lattice are relatively inadequate [64,65,66]. For example, high-resolution transmission electron microscopy (HR-TEM) analysis of rose CDs fails to discern well-defined graphite lattice patterns (with lattice spacings of 0.246 nm or 0.335 nm), and X-ray diffraction (XRD) data is lacking for the validation of the graphite lattice (with 2θ values of 18.2° or 23.8°) [64]. Additionally, the elemental composition of CDs derived from *Rhei radix* rhizome and *Phellodendri chinensis* cortex reveals an excessively high proportion of oxygen elements (24.2% and 28.4%), implying that the synthesis of these CDs may have undergone excessive oxidation [65,66]. Ideal synthesis conditions for these CDs still require further optimization, implying potential progressiveness in the biomedical applications of CDs produced through powder carbonization.

Hydrothermal carbonization is another method that has been used to produce CDs from raw foods or edible plants [71]. This method involves immersing the substance in a solvent, such as pure water, HCl, NaOH, or EtOH. The hydrothermal carbonization reaction is commonly enclosed in a Teflon autoclave, enabling a prolonged heating procedure at relatively high temperatures, typically ranging from 180–500 °C, for a duration of 2–12 h [71]. The pristine material in the solvent undergoes uniform heating, providing the necessary energy to facilitate oxidation, dehydration, cross-linking, and polymerization of functional groups [23,67,71]. This results in the formation of a carbon core predominantly based on the sp^2^ structure, such as graphene [68]. The hydrothermal carbon conversion process is gentler compared to dry burning, reducing the likelihood of excessive carbonization and retaining a higher degree of biologically functional groups on the surface of CDs [23,67,71]. Considering the main requirement of biosafety, pure water is used as the solvent for hydrothermal carbonization in food-derived CDs. Furthermore, microwave-assisted hydrothermal carbonization (MWAHTC) is another method that can be used to generate CDs from raw food or edible plant materials [72]. This method utilizes high microwave power to induce rapid internal vibrations in water molecules within the material, allowing for shorter processing times (5–60 min) to achieve efficient thermal energy transfer. The additional energy supplied is typically measured in watts (W), with power levels typically ranging from 70 to 1000 W. MWAHTC enables the conversion of carbon materials in a state that closely resembles the original food, and the common homogenization strategy involves simple chopping or grinding [72].

In comparison to conventional food processing methods, researchers found that dry burning, hydrothermal treatment, and microwave heating are more efficient in altering the properties of food by carbonizing its nutrient content into CDs [23]. In previous studies, researchers have used hydrothermal treatment or microwave heating processes in a wide variety of foods, including milk [73,74,75], fruits [76,77,78,79,80,81,82], and vegetables [83,84,85,86,87,88,89,90,91,92,93,94,95,96,97,98,99,100,101,102,103,104,105,106,107,108,109,110,111,112,113] to produce CDs. These treatments have led to the production of CDs with sizes ranging from 0.5 to 31.1 nm (Table 2). Other studies have used natural flavor enhancers or natural sweeteners, such as guar gum and honey, for synthesizing CDs using MWAHTC or hydrothermal methods [112,113]. The use of the hydrothermal method has also been demonstrated in the production of CDs from probiotics that regulate intestinal flora to promote nutrient digestion/absorption and enhance immunity, including *Escherichia coli*, *Bifidobacterium breve*, *Nannochloropsis oculata*, and *Bacillus cereus* [114,115,116]. The size of the CDs generated from these probiotics was reported to range from 1.0 to 9.3 nm [114,115,116].

Notably, certain edible plants such as *Phellodendri chinensis* [65], *Rhei Radix* [66], forsythia [86], Chinese mugwort [88], ginkgo [90], green chiretta [91], Henna [92], rosemary [95], tea tree [97,98], ginger [99], and turmeric [103,104] are renowned for their therapeutic effects. These plants have gained popularity as valuable sources of carbon in the quest for CDs with medical properties (Table 2). Traditional Chinese medicine utilizes specific parts of these edible plants, including peels, fruits, seeds, roots, stems, and leaves, as raw materials [40]. These parts are collected and exposed to prolonged sunlight before being ground into powder. To transform them into therapeutic drugs, these medicinal plants undergo rigorous testing and processing, which includes heat treatment. Due to their richness in bioactive substances such as polysaccharides, polyphenols, and terpenoids, these plants constitute valuable resources for medicinal properties [117]. Interestingly, the process of transforming natural ingredients into CDs through heat treatment bears some resemblance to the traditional preparation of Chinese herbal medicine, where heat treatment is applied to raw materials [40,117]. Note that the manufacturing process of CDs employs modern molecular cooking techniques, utilizing high-purity molecules and precise processing conditions [40].

**Table 2 ijms-24-16579-t002:** Examples of CDs synthesized from raw food or edible plants.

Food Groups	Food Source	Synthetic Method	Types/Size (nm)	Quantum Yield (%)	Toxic Evaluation	Potential Biomedical Applications	Ref.
Raw meat	Lamb	Oven heating (280 °C for 15–45 min)	CDs/2.6–4.1	10	**In vitro** >4 mg/mL, 24 h (PCl12 cells)	Protein adsorption	[26]
	Lamb	Oven heating (200–300 °C for 30 min), then extraction by ethanol for 24 h	CDs/1.7–2.8	6–45	**In vitro** >2 mg/mL, 7 h (HepG2 cells)	Scavenging ROS	[27]
	Beef	Oven heating (280 °C for 30 min), then extraction by ethanol for 30 min	CDs/1.0–4.0	NA	**In vitro** >1 mg/mL, 12 h (NRK cells)	Protein adsorption	[28]
	Beef broth	Oven heating (117 °C for 30–70 min), then extraction by ethanol for 40 min	CNPs/2.4–5.4	2.0–2.5	**In vitro** >10 mg/mL, 24 h (NRK cells)	Carrier for zinc	[29]
	Duck	Oven heating (200–300 °C for 30 min), then extraction by ethanol for 1 h	CDs/1.5–3.2	10.5–38.0	**In vitro** >4 mg/mL, 36 h (PC12 cells) **In vivo** >15 mg/mL, 24 h (*C. elegans*)	In vivo *C. elegans* bio-imaging	[30]
	Duck	Oven heating (170 °C for 1 h) then extract by ethanol for 1 h	CNPs/0.7–2.3	4.4	NA	Protein adsorption	[31]
	Chicken	Oven heating (150–300 °C for 1 h) then extraction by ethanol for 36 h	CDs/1.5–20.4	6.5–17.9	**In vitro** >4 mg/mL, 24 h (HepG2 cells) **In vivo** >2 g/kg, 20 h (mice)	Dopamine sensing	[32]
	Pike eel	Oven heating (160–300 °C for 30 min), then extraction by ethanol for 24 h	CNs/1.8–4.3	80.2	**In vitro** >20 mg/mL, 24 h (MC3T3-E1 cells)	In vitro bio-imaging	[33]
	Atlantic salmon	Oven heating (200 °C for 10–60 min), then extraction by ethanol for 2 h	CQDs/1.9–4.1	2.2–12.1	**In vitro** >6 mg/mL, 6 h (NRK cells) **In vivo** >2 g/kg, 24 h (mice)	In vivo mice bio-imaging	[34]
	Mackerel	Oven heating (230 °C for 40 min), then extraction by ethanol for 2 h	CDs/0.9–3.5	12.0	NA	Scavenging ROS	[35]
	Spanish Mackerel	Grill-heating (230 °C for 30 min) and then extraction by 10% methanol for 2 h	CDs/2.9–3.0	NA	NA	Protein adsorption	[36]
Processed food	Breadcrumbs	Oven heating (180 °C with cooking oil) then extraction by petroleum ether for overnight	CDs/2.6–4.0	1.0	NA	Protein adsorption	[48]
Flavor enhancers	Grounded spice of cinnamon, red chili, turmeric and black pepper	Hydrothermal (200 °C for 12 h)	CDs/10.3–15.0	NA	**In vitro** >2.0 mg/mL, 24 h (HK2 cells)	In vitro bio-imaging/Anticancer	[118]
Milk	Commercial cow milk	Hydrothermal (190–200 °C for 1–8 h)	CDs/ 0.5–4.0	NA	**In vitro** >0.4 mg/mL, 24 h (HT22 cells)	Scavenging ROS	[73]
	Commercial fat-free cow milk	Hydrothermal (180 °C for 2 h)	CDs/ 2.0–4.0	12	**In vitro** >1 mg/mL, 24 h (U87 cells)	In vitro bio-imaging	[74]
	Cow yogurt	Microwave (800 W for 30 min)	CDs/1.4–9.5	1.5	**In vitro** >7.1 mg/mL, 100 h (MCF-7 and CoN cells)	In vitro bio-imaging	[75]
Fruits	Kiwi, Avocado, or Pear	Hydrothermal (200 °C for 12 h)	CDs/4.0–4.5	20–35	**In vitro** >1.2 mg/mL, 72 h (HK-2 cells)/>2.2 mg/mL, 72 h (Caco-2 cells) **In vivo** >64 mg/mL, 80 h (zebra fish embryo)	In vivo zebrafish bio-imaging/Anticancer	[76]
	Mango	Hydrothermal (100 °C for 1 h, in H_2_SO_4_; 80 °C for 15 min, in H_3_PO_4_; 80 °C for 30 min, in H_3_PO_4_), then adjusted to pH 7.0 with NaOH	CNPs/5.0–10.0, 5.0–10.0, or 10.0–14.0	3.9, 1.6, or 0.5	**In vitro** >5 mg/mL, 24 h (A549 cells) **In vivo** >5 mg/kg, 24 h (mice)	In vivo mice bio-imaging	[77]
	Sapodilla fruits	Hydrothermal (100 °C for 1 h, in H_2_SO_4_; 80 °C for 15 min, in H_3_PO_4_; 80 °C for 30 min, in H_3_PO_4_), then adjusted to pH 7.0 with NaOH	CDs/1.6–2.2, 2.2–3.6, or 3.3–5.8	5.7, 7.9, or 5.2	**In vitro** >300 µg/mL, 15 h (HeLa cells)	In vivo bacterial/Fungal bio-imaging	[78]
	Cherry plum juice	Hydrothermal (200 °C for 20 h)	CDs/1.0–8.0	NA	**In vitro** >500 µg/mL, 24 h (HepG2 cells)	In vitro bio-imaging	[79]
	Lemon juice	Hydrothermal (120 °C for 3 h)	CQDs/2.0–4.5	9.0	NA	In vivo plant bio-imaging (onion epidermal cells)	[80]
	Tomato juice	Hydrothermal (160 °C for 3 h)	CDs/2.4–3.6	NA	**In vitro** >100 µg/mL, 96 h (A549, and Human dermal fibroblasts cells)	Scavenging ROS	[81]
	Watermelon juice/Orange juice/Lemon juice/Cantaloupe juice/Red plum juice/Green plum juice/Carrot juice/Red pitaya juice/White pitaya juice	Hydrothermal (180 °C for 4 h)	CDs/1.6–5.6	13–25	**In vitro** >1 mg/mL, 4 h (RAW 264.7 cells) **In vivo** >2 mg/kg, 5 h (zebrafish eleutheroembryo)/3.2 mg/kg, 6 h (zebrafish eleuthero-embryo)	In vivo zebrafish bio-imaging (ROS sensing)	[82]
Edible plants	Linseed (seeds)	Hydrothermal (180 °C for 12 h)	CDs/4.0–8.0	14.2	**In vitro** >200 µg/mL, 24 h (MCF-7 cells)	In vitro bio-imaging	[83]
	Peanuts (seeds)	Hydrothermal (250 °C for 6 h)	CDs/2.0–8.0	7.9	**In vitro** >1 mg/mL, 24 h (MCF-7 cells)	In vitro bio-imaging	[84]
	Wheat bran (seeds)	Hydrothermal (180 °C for 3 h)	CDs/ca. 4.9	33.2	**In vitro** >6 mg/mL, 24 h (SH-SY5Y cells)	Drug carrier (amoxicillin; antibiotic)	[85]
	Forsythia (dried fruit powder) + Urea + Ethanolamine	Microwave (300 W for 2 min, repeat 3 times)	CQDs/1.8–3.6	NA	NA	Antifungal	[86]
	Rose (flower petals) + thymol	Powder carbonization (180 °C for 6 h) and decorate with thymol	CDs/5.0–6.0	NA	**In vivo** >10 mg/kg, 144 h (rats)	Immuno-modulatory effect	[64]
	*Phellodendri chinensis* (Cortex)	Powder carbonization (400 °C for 1 h)	CDs/0.5–3.6	5.6	**In vitro** >39 µg/mL, 24 h (L02, 293T, and RAW 264.7 cells) **In vivo** >0.86 mg/kg, 7 days (mice)	Immuno-modulatory effect	[65]
	Cabbage (leaves)	Hydrothermal (140 °C for 5 h)	CQDs/2.0–8.0	16.5	**In vitro** >700 µg/mL, 24 h (HaCaT cells)	In vitro bio-imaging	[87]
	Chinese mugwort (leaves)	Purified fume particulate matter	CDs/3.0–7.0	NA	**In vitro** >150 µg/mL, 24 h (HEK 293T cells)	Antibacterial	[88]
	Coriander (leaves)	Hydrothermal (240 °C for 4 h)	CDs/1.5–3.0	6.48	**In vitro** >1 mg/mL, 12 h (A549 and L-132 cells)	Scavenging ROS/In vitro bio-imaging	[89]
	Ginkgo (leaves)	Hydrothermal (200 °C for 10 h)	CQDs/2.0–4.0	22.8	NA	Disease detection in mouse serum	[90]
	Green chiretta (leaf extract)	Hydrothermal (160 °C for 8 h)	CDs/8.0–11.0	15.1	**In vitro** >700 µg/mL, 24 h (MCF-7 cells)	Scavenging ROS/In vitro bio-imaging/Antibacterial/Anticancer	[91]
	Henna (leaves)	Hydrothermal (180 °C for 12 h)	CDs/2.7–7.8	28.7 (Rhodamine B)	NA	Antibacterial/Anticancer drug sensing	[92]
	Holy basil (leaves)	Hydrothermal (180 °C for 4 h)	CDs/1.0–4.0	9.3	**In vitro** >200 mg/mL, 24 h (MDA-MB-648 cells)	In vitro bio-imaging	[93]
	Pakchoi (leaves)	Hydrothermal (150 °C for 12 h)	CDs/1.0–3.0	37.5	**In vitro** >2 mg/mL, 24 h (HeLa cells)	In vitro bio-imaging	[94]
	Rosemary (leaves)	Hydrothermal (140–200 °C for 6–12 h)	CDs/11.5–20.7	NA	NA	Antibacterial	[95]
	Spinach (leaves)	Hydrothermal (150 °C for 6 h)	CDs/3.0–11.0	15.3	**In vitro** >200 µg/mL, 24 h (A549 cells) **In vivo** >2 mg/mL, 24 h (mice)	In vivo tumor imaging in mice	[96]
	Tea tree (leaves)	Hydrothermal (220 °C for 3 h)	CDs/1.7–5.0	4.9	**In vitro** >4 mg/mL, 24 h (HepG2 cells)	In vitro bio-imaging	[97]
	Tea tree /Osmanthus/Milk vetch (leaves)	Hydrothermal (200 °C for 2 h)	CDs/3.0–18.0	NA	**In vitro** >1 mg/mL, 24 h (293T cells)	Antibacterial	[98]
	Escallion (stem)	Hydrothermal (220 °C for 3 h)	CDs/ca. 4.22	10.5	**In vitro** >200 µg/mL, 24 h (MCF-7 and K562 cells)	In vitro bio-imaging	[99]
	Garlic (bulb)	Hydrothermal (180 °C for 10 h)	CDs/ca. 3.6	6.8	NA	In vitro bio-imaging	[100]
	Ginger (rhizome)	Hydrothermal (300 °C for 20 min)	CDs/3.5–5.1	13.4	**In vitro** >2.8 mg/mL, 24 h (A549, MDA-MB-231, and FL83B cells)/>1.4 mg/mL, 24 h (HeLa cells)/>0.4 mg/mL, 24 h (HepG2 cells)	Anticancer	[101]
	Konjac (bulb)	Powder carbonization (470 °C for 1.5 h)	CDs/ca. 3.4	13.0	**In vitro** >150 mg/mL, 12 h (HeLa cells)	In vitro bio-imaging	[102]
	*Rhei radix* (rhizome)	Powder carbonization (350 °C for 1 h)	CDs/1.4–4.5	NA	**In vitro** >200 µg/mL, 24 h (RAW 264.7 cells)	Immuno-modulatory effect	[66]
	Turmeric (rhizome)	Hydrothermal (180 °C for 10 h)	CDs/1.5–4.0	NA	**In vitro** >200 µg/mL, 24 h (PC3 cells)	Antibacterial effects	[103]
	Turmeric (rhizome) + Ammonium persulfate	Hydrothermal (200 °C for 6 h)	CDs/9.4–11.8	NA	**In vitro** >1 mg/mL, 72 h (L929 cells)	Antibacterial effects/Scavenging ROS	[104]
	Yam (stem tuber)	Hydrothermal (200 °C for 2 h)	CDs/1.5–4.0	9.3	NA	Anticancer drug sensing	[105]
	Beetroot (root)	Hydrothermal (160 °C for 8 h)	CDs/<5.0	11.6	**In vitro** >2.5 µg/mL, 24 h (HEK-293 cells)	Anticancer/Scavenging ROS	[106]
	Carrot (root)	Hydrothermal (170 °C for 12 h)	CDs/ca. 2.3	7.6	**In vitro** >2 mg/mL, 24 h (MCF-7 cells)	Drug carrier (mitomycin; anticancer)	[107]
	Rose-heart radish (root)	Hydrothermal (180 °C for 3 h)	CDs/1.2–6.0	13.6	**In vitro** >500 µg/mL, 3 h (SiHa cells)	In vitro bio-imaging	[108]
	Sweet potato (root)	Hydrothermal (180 °C for 18 h)	CDs/2.5–5.5	8.6	**In vitro** >150 µg/mL, 24 h (HeLa, HepG2 cells)	In vitro bio-imaging	[109]
	Oyster mushroom (Sporocarp)	Hydrothermal (120 °C for 4 h; dissolved in 5% H_2_SO_4_)	CDs/5.0–18.0	NA	**In vitro** >25 µg/mL, 24 h (HEK 293 cells)	Antibacterial/Anticancer	[110]
	Water chestnut (bulb) + Onion (bulb)	Hydrothermal (180 °C for 4 h)	CDs/2.0–4.0	12.0	**In vitro** >300 µg/mL, 24 h (T24 cells)	In vivo bio-imaging and quantification of coenzyme A (pig liver)	[111]
Natural flavor enhancers	Guar gum (Seed endosperm)	Microwave (400 W for 30 min)	CDs/19.2–31.1	7.5	**In vivo** >1 mg/mL, 1 h (China rose leaf)	In vivo plant bio-imaging (China rose leaf guard cells)	[112]
	Honey + Garlic (bulb)+ Ammonia	Hydrothermal (200 °C for 6 h)	CQDs/4.0–13.0	4.2	NA	Antibacterial	[113]

NA—not available; CDs—carbon dots; CNPs—carbon nanoparticles; CNs—carbonaceous nanostructures; CQDs—carbon quantum dots.

### 2.3. CDs Synthesized from Dietary Compounds

Several studies have explored various carbonization strategies to obtain CDs from numerous edible compounds, such as food additives for food processing (Appendix A) or health-promoting dietary compounds (Table 3) [119,120,121,122,123,124,125,126,127,128,129,130,131,132,133,134,135,136,137,138,139,140,141,142,143,144,145,146,147,148,149,150,151]. For example, CDs from alginate, curcumin, folicate, hesperidin, spermidine, and quercetin were obtained through a dry burning process (300–500 °C; 20–300 min) [132,133,134,135,136,137,138,139,140,141]. Another method, the so-called microwave method, has also been used to achieve the carbonization of these compounds in a shorter duration [72]. This method has been used to synthesize CDs from amino acids, proteins (i.e., casein and RNase A), and chitosan [132,133,134,135,136,137,138,139,140,141]. In other studies, researchers employed the hydrothermal method to enable the thermal carbon conversion of caffeic acid, fucoidan, and glycyrrhizic acid, resulting in the carbonization of CDs [142,143,144,145,146,147,148,149,150,151].

The identification of CDs derived from dietary compounds is typically performed using various analytical techniques, including Fourier-transform infrared spectroscopy (FT-IR), XRD, X-ray photoelectron spectroscopy (XPS), liquid chromatography-mass spectrometry (LC-Mass) and nuclear magnetic resonance spectroscopy (NMR) [152]. Several studies have revealed that utilizing dietary compounds in the carbonization reaction enables the inheritance of functional groups onto the surface of the carbon core throughout the heating process, resulting in the formation of novel CDs [119,120,121,122,123,124,125,126,127,128,129,130,131,132,133,134,135,136,137,138,139,140,141,142,143,144,145,146,147,148,149,150,151]. These functional groups, inherited to the CD surface, may then undergo thermal activation, leading to processes such as residue loss (e.g., decarboxylation and dehydrogenation), rearrangement (e.g., keto-enol tautomerism and conformational changes in aromatic rings), or fusion (e.g., carboxyl fusion and carboxylic acid-amine cross-reaction) [67,68]. The combinations of these newly generated functional groups during the carbonization process could potentially exhibit enhanced biological activities compared to the original bioactive compounds [119,120,121,122,123,124,125,126,127,128,129,130,131,132,133,134,135,136,137,138,139,140,141,142,143,144,145,146,147,148,149,150,151]. For example, Mao et al. demonstrated that the phenolic-like functional groups were produced during the carbonization reaction of alginate, which was not present in the pristine material [18].

Moreover, the carbonization process can modify the inherent physical properties of the resulting products by inheriting functional groups [20,21,119,130]. For instance, the one-step carbonization synthesis of curcumin, quercetin, sorbitan monolaurate (a common hydrophobic compound used as a food emulsifier), and triolein (main components in cooking oil) into CQDs [119], CNGs [130], CNVs [21], or CD_somes_ [20] was found to increase water dispersibility significantly. Additionally, combining with soluble components has demonstrated comparable effects. As an illustration, the co-carbonization of quercin and lysine notably improves the water solubility and biocompatibility of the resulting Qu/Lys-CNGs [17]. These enhancements in water dispersibility can be attributed to the distinctive combination of functional groups generated through thermal activation, leading to increased hydrophilicity and facilitating interactions with water molecules [17,119,130] or the formation of specific structures like nanovesicles [19,20,21,22]. These nanovesicles play a crucial role in the development of vesicular structures via amphiphilic CDs, with the hydrophilic end located on the outermost side of the CDs (i.e., CNVs and CD_somes_).

**Table 3 ijms-24-16579-t003:** Examples of CDs synthesized from active compounds.

Precursor	Synthetic Method	Type/Size (nm)	Yield (%)	Quantum Yield (%)	Toxic Evaluation	Potential Biomedical Applications	Ref.
Ammonium citrate/Spermidine	Powder carbonization (180 °C for 2 h and 260 °C for 2 h)	CDs/3.8–5.4	50.8	2.8	**In vitro** >50 mg/mL, 24 h (HEK-293T, MCF-7, A549, HeLa, and HaCaT cells) **In vivo** >50 mg/mL, 12 days (mice)	Antibacterial/Wound healing	[153]
Citric acid + Diethyl-enetriamine	Powder carbonization (170 °C for 3 h in a nitrogen atmosphere)	CDs/5.0–8.0	NA	25.5	**In vitro** >100 μM, 24 h (A2780 cells) **In vivo** >100 μM, 14 days (mice)	In vivo tumor image in mice/Drug carrier (cisplatin; anticancer)	[154]
Curcumin	Powder carbonization (180 °C for 2 h)	CQDs/ 4.2–5.2	10.0–25.0 (*w*/*w*)	0.3	**In vitro** >50 mg/mL, 24 h (RD cells) **In vivo** >25 mg/kg, 15 days (mice)	Antivirus	[119]
Curcumin	Powder carbonization (180 °C for 2 h)	CQDs/ ca. 4.8	NA	NA	**In vitro** >100 mg/mL, 24 h (BHK-21 cells)	Antivirus	[120]
Folic acid	Powder carbonization (140 °C for 6 h)	CDs/1.0–1.6	NA	NA	**In vitro** >200 μg/mL, 72 h (chondrocytes and macrophages) **In vivo** >2 mg/kg, 6 weeks (mice)	Immuno-modulatory	[121]
Glutamic acid	Powder carbonization (210 °C for ~1 min)	GQDs/3.4–5.9	NA	54.5 (NaOH)	**In vitro** >10 mg/mL, 1 h (MH-S cells) **In vivo** >25 mg/mL, 1 h (mice)	In vivo bioimage in mice	[122]
Hesperidin	Powder carbonization (250 °C for 2 h)	CPDs/46.7–60.1	NA	NA	**In vitro** >500 μg/mL, 72 h (RD cells) **In vivo** >25 mg/kg, 9 days (mice)	Antivirus	[123]
Spermidine	Powder carbonization (270 °C for 3 h)	CQDs/ ca. 6.0	NA	2.0–4.3	**In vitro** >200 mg/mL, 24 h (RCK cells)	Antibacterial	[124]
Spermidine	Powder carbonization (270 °C for 3 h)	CQDs/ ca. 6.0	NA	2.0–4.3	**In vitro** >200 mg/mL, 24 h (RCK cells)	Antivirus	[125]
Spermine + Dopamine	Powder carbonization (250 °C for 2 h)	CQDs/ ca. 10.0	11.4	4.3	**In vitro** >100 μg/mL 24 h (SIRC cells) **In vivo** >200 μg/mL, 14 days (rabbit)	Antibacterial	[155]
Citric acid/boronic acids	Powder carbonization (250 °C for 0.5 h) and then mix with the boronic acid solution	CQDs/5.4–7.0	N.A	N.A	**In vitro** >600 μg/mL, 24 h (MOLT-4 cells)	Antivirus	[156]
Lysine	Powder carbonization (270 °C for 3 h)	CNGs/120.0–510.0	66.5	8.1	**In vitro** >50 μg/mL, 24 h (BHK-21 and Vero cells) **In vivo** >30 μg/mL, 7 days (chicken embryo)	Antivirus	[126]
Lysine	Powder carbonization (270 °C for 3 h)	CNGs/118.9–178.7	66.5	8.1	**In vitro** >100 μg/mL, 24 h (HUVEC, RD, HepG2, HaCaT, and HEK-293T cells)	Antibacterial	[127]
Lysine	Powder carbonization (270 °C for 3 h)	CNGs/118.9–178.7	66.5	8.1	**In vivo** 50 μg/mL, 96 h (zebrafish embryos)/10 μg/mL, 96 h (zebrafish eleutheroembryo)/0.5 μg/mL, 90 days (adult zebrafish)/2000 mg/kg, 48 h (guinea pigs)/2000 mg/kg, 72 h (rabbit)/2000 mg/kg, 14 days (rats)	In vivo bioimage in zebrafish	[128]
Lysine or Arginine	Powder carbonization (240 °C for 3 h)	CQDs/2.0–7.0	NA	NA	**In vitro** >1 mg/mL, 24 h (NIH-3T3, BMSCs, and HUVECs cells) **In vivo** >2 mg/mL, 5 days (mice)	Antibacterial/Scavenging ROS/Promoting tissue repair in mice	[129]
Quercetin	Powder carbonization (270 °C for 2 h) then dissolved in sodium phosphate buffer (pH 12)	CNGs/326.9–423.3	78	<1	**In vitro** >1 mg/mL, 24 h (MDCK cells) **In vivo** >500 μg/mL, 14 days (mice)	Antivirus	[130]
Quercetin + Lysine	Powder carbonization (270 °C for 3 h)	CNGs/44.8–235.2	17.5	3.3	**In vitro** >100 μg/mL, 24 h (SIRC cells) **In vivo** >50 μg/mL, 28 days (rabbit)	Antibacterial/Scavenging ROS/Anti-inflammatory effects	[17]
Sodium alginate + Ammonium sulfite	Powder carbonization (180 °C for 3 h)	CNGs/116.0–183.0	31.2	13.0	**In vitro** >1 mg/mL, 24 h (MDCK cells) **In vivo** >500 μg/mL, 14 days (mice)	Antivirus/Scavenging ROS/Anti-inflammatory effects	[157]
Sorbitan monolaurate	Powder carbonization (230 °C for 3 h) then dissolved in ethanol	VCDs/390–430	NA	NA	NA	Enzyme and nanomaterial carrier/Cholesterol detection in serum	[21]
Asparagine	Microwave (180 °C for 15 min)	CDs/ca. 1.4	NA	<1	**In vitro** >800 μg/mL, 24 h (HeLa cells)	In vitro bioimage	[131]
Casein (milk protein)	Microwave (450 W for 30 min; heating for 2 min and then pausing for 15 s)	CDs/ca. 1.6	NA	18.7	**In vivo** >200 μg/mL, 10 min (spinach leaf)	In vivo plant bio-imaging (spinach guard and epidermal cells)	[132]
Chitosan	Microwave (700 W for 9.5 min)	CDs/2.7–6.5	6.4	6.4	NA	In vitro bioimage	[133]
Citric acid + Cysteine	Microwave (140 °C for 25 min)	CQDs/0.9–1.0	NA	91.2	**In vivo** ca. 1 mL/mice, 3 h (mice)	Drug carrier (insulin)/In vivo glycemic control	[134]
Citric acid + Poly-ethyleneimine	Microwave (1150 W for 3 min) then mixed with locked nucleic acid (LNA)	CDs/ca. 3.7	NA	NA	**In vitro** >1 μg/mL, 3 days (KMM, BC3, BCP1, BCBL1, and BJAB cells) **In vivo** >50 μg/mice, 3 weeks (mice)	Antivirus	[135]
Citric acid + RNase A enzyme	Microwave (700 W for 3–5 min)	CDs/ca. 4.0	NA	24.2	**In vitro** >3 mg/mL, 24 h (MGC-803 cells) **In vivo** >5 mg/mL, 24 h (mice)	In vivo tumor imaging in mice	[136]
Citric acid + Tryptophan	Microwave (700 W for 3 min)	CDs/ca. 2.6	NA	20.6	**In vitro** >400 μg/mL, 24 h (MGC-803 cells)	In vitro bioimage/Drug carrier (siRNA)	[137]
Citric acid + Urea	Microwave (800 W for 15 min)	CDs/1.0–5.5	NA	NA	NA	Antibacterial	[138]
Citric acid + Urea	SPMA (6 kW for 5 min)	GQDs/ 3.0–20.0	ca. 40	NA	**In vitro** >50 μg/mL, 72 h (H171 cells)	Antivirus	[139]
Citric acid + Urea	Microwave (650 W for 4–5 min), then powder carbonization (60 °C for 1 h)	CDs/2.0–6.0	NA	36.0	**In vitro** >100 μg/mL, 96 h, (HepG2 and HL-7702 cells) **In vivo** >500 μg/mL, 14 days (mice)	Drug carrier (doxorubicin; anticancer)/In vivo tumor imaging in mice	[140]
Glucose + Arginine	Microwave (700 W for 10 min)	CDs/1.0–7.0	NA	12.7	**In vitro** >200 μg/mL, 24 h (MEFs cells)	In vitro bioimage/Drug carrier (circular DNA)/Chondrogenic differentiation	[141]
Microcrystalline cellulose	Alkaline hydrolysis (90 °C for 2 h), then infrared-assisted heating (125 °C for 6 h)	CQDs/6.7–12.5	NA	NA	NA	Antibacterial/Anticancer	[148]
Boronic acid derivatives	Hydrothermal (160 °C for 8 h)	CQDs/8.9–9.5	NA	0.05	**In vitro** >100 μg/mL, 8 h (Huh-7 cells)	Antivirus	[158]
Ciprofloxacin (antibiotic)	Hydrothermal (200 °C for 4 h)	CDs/4.7–6.8	NA	25.3	NA	Antibacterial	[159]
Citric acid + amino acid (Arg, Cys, Glu, Gly, His, Leu, Phe, and Tyr)	Hydrothermal (180 °C for 12 h; dissolved in formamide)	CDs/3.0–6.0	NA	25.5–62.1	**In vitro** >100 μg/mL, 24 h (HeLa cells)	In vitro bioimage	[160]
Citric acid + Curcumin	Hydrothermal (180 °C for 1 h)	CQDs/1.2–1.8	NA	3.6	**In vitro** >250 μg/mL, 18 h (RAW 264.7 cells)	Antivirus	[161]
Citric acid + Branched poly-ethyleneimine	Hydrothermal (200 °C for 12 h)	CQDs/2.0–8.0	NA	NA	**In vitro** >500 μg/mL, 72 h (L929 cells)	Antibacterial	[162]
Citric acid + Curcumin	Hydrothermal (180 °C for 24 h)	CDs/1.5–2.5	NA	30	**In vitro** >250 μg/mL, 48 h (RAW 264.7, HK-2, and HPMCs cells)	Antibacterial	[163]
Citric acid + Ethyl-enediamine/ampicillin (antibiotic)	Hydrothermal (250 °C for 4 h) coupled with ampicillin conjugation	CDs/ca. 34.0–54.0	60	19	**In vitro** >200 μg/mL, 24 h (HeLa cells)	Antibacterial	[164]
Vit C + PEG-diamine	Hydrothermal (180 °C for 1 h)	CDs/4.7	NA	NA	**In vitro** >250 μg/mL, 48 h (PK-15 and MARC-145 cells)	Antivirus	[165]
Caffeic acid	Hydrothermal (200 °C for 6 h)	CQDs/1.5–2.5	10.2	NA	**In vitro** >10 mg/mL, 12 h (HeLa cells)	Antibacterial/Antivirus	[142]
Carrageenan or Pullulan	Alkaline hydrolysis (90 °C for 2 h), then hydrothermal (210 °C for 6 h)	CQDs/ ca. 3.1 or ca. 4.2	NA	NA	**In vitro** >1000 or >500 μg/mL, 24 h (Vero E6 cells)	Antivirus/Anticancer	[143]
Chlorogenic acid + Caffeic acid + Quinic acid	Hydrothermal (230 °C for 2 h)	CQDs/5.0–10.0	NA	NA	**In vitro** >100 μg/mL, 24 h (L02 cells) **In vivo** >200 mg/kg, 90 min (mice)	Anticancer/GSH oxidase-like activity/Scavenging ROS	[52]
Folic acid	Hydrothermal (180 °C for 2 h)	CDs/3.0–11.0	NA	23.0	**In vitro** >1 mg/mL, 3 h (U87 cells)	In vitro bioimage	[144]
Fucoidan	Hydrothermal (200 °C for 12 h)	CDs/4.0–10.0	NA	NA	**In vitro** >1 mg/mL, 3 h (MC3T3-E1 cells)	Antibacterial	[145]
Glucose, Vit C, or Fructose	Hydrothermal (200 °C for 12 h)	CDs/ca. 9.0–10.0	34/56/29 (*w*/*w*)	1.8/1.5/0.3	**In vitro** >1000/>250/<1 μg/mL, 96 h (HeLa cells)	Drug carrier (doxorubicin)	[146]
Glucose + Ethylenediamine	Hydrothermal (200 °C for 4 h)	CDs/1.0–3.0	NA	NA	**In vivo** >2.5 mg/mL, 3 h (zebrafish embryos)/>1.5 mg/mL, 10 h (zebrafish eleuthero-embryos)	In vivo bio-imaging in zebrafish embryos and eleuthero-embryos	[166]
Glucose + Glutamic acid	Hydrothermal (125 °C for 30 min, then 200 °C for 20 min; dissolved in NaOH)	CDs/ca. 2.0	29.8	NA	**In vitro** >1000 μg/mL, 48 h (HeLa cells)	Drug and fluorescent dye carrier (doxorubicin; anticancer)/In vitro bioimage	[167]
Glucose + Aspartic acid	Hydrothermal (125 °C for 30 min, then 200 °C for 20 min; dissolved in NaOH)	CDs/1.8–2.7	34.5	7.5	**In vitro** >500 μg/mL, 48 h (L929 and C6 cells) **In vivo** >200 mg/kg, 90 min (mice)	In vivo tumor image in mice	[168]
Glycyrrhizic acid	Hydrothermal (180 °C for 7 h; dissolved in NaOH)	CQDs/ ca. 11.4	NA	1.4	**In vitro** >450 μg/mL, 48 h (MRC 145 cells) **In vivo** >200 mg/kg, 90 min (mice)	Antivirus	[143]
Sorbitol + Ethyl-enediamine	Hydrothermal (180 °C for 5 h)	CDs/ca. 5.0	NA	8.9	**In vitro** >1000 μg/mL, 24 h (MCF-7 cells)	In vitro bioimage	[160]
Vitamin C	Hydrothermal (180 °C for 4 h)	CDs/ca. 9.0	NA	NA	**In vitro** >1 mg/mL, 48 h (NIH-3T3 cells) **In vivo** >1 mg/mL, 48 h (fungus)	Fluorescent dye carrier/In vivo bioimaging in fungus *Candida albicans*	[149]
Triolein	Hydrothermal (220 °C for 3 days), then dissolved in NaOH	CD_somes_/80.0–100.0	ca. 30	4.1	**In vitro** >300 μg/mL, 24 h (HaCaT cells) **In vivo** >100 μg/mL, 12 days (mice)	Antibacterial/Controllable ROS induction/Wound healing	[150]
Triolein	Hydrothermal (220 °C for 3 days), then dissolved in NaOH	CD_somes_/80.0–100.0	NA	1.0	**In vitro** >300 μg/mL, 48 h (HeLa cells)	In vitro bioimage	[20]
Triolein	Hydrothermal (220 °C for 3 days), then dissolved in NaOH	CD_somes_/80.0–100.0	68	NA	**In vitro** >400 μg/mL, 24 h (NIH-3T3 cells)	Anticancer/Controlable ROS induction	[151]
Citric acid + Glutathione	Oil bath (200 °C)	CDs/2.5–3.0	NA	80.3	**In vitro** >3 mg/mL, 24 h (A549 cells)	In vitro bioimage	[169]
Vitamin C	Electrolysis (0.1 A for 3 weeks)	CDs/3.0–6.0	NA	ca. 30	NA	Antibacterial/Antifungal	[170]

NA—not available; CDs—carbon dots; CD_somes_—carbon dot liposomes; CNGs—carbon nanogels; CNPs—carbon nanoparticles; CPDs—carbon polymer dots; CQDs—carbon quantum dots; GQDs—graphene quantum dots; VCDs—vesicle-like carbon dots. SPMA—solid-phase microwave-assisted technique.

## 3. Biomedical Applications of Food-Based CDs

### 3.1. Bio-Imaging Applications

The cellular uptake of nanomaterials occurs through diverse pathways, broadly categorized as dynamin-dependent endocytosis and passive diffusion [171]. In the case of smaller nanomaterials (<10 nm), passive diffusion allows them to directly translocate to the cytoplasm for entry, requiring no energy consumption and are driven by concentration gradients [172]. From a biomolecular standpoint, endocytosis-based uptake pathways are regulated and mediated by various lipids and transporters (such as clathrin, caveolin, lipid rafts, dynamin, actin, and pattern recognition receptors) [171]. Nanomaterials in the size range of 100–500 nm, upon dynamin-dependent endocytosis, are initially recognized by cell receptors, leading to membrane curvature and, subsequently, the formation of intracellular vesicles (e.g., endosomes, phagosomes, or macropinosomes) through the activation of GTPase enzyme known as dynamin [172]. Subsequently, these intracellular vesicles undergo lysosomal acidification to facilitate the release of nanomaterials. Hence, nanomaterials can serve as carriers for drugs. This is particularly important because CDs are increasingly recognized for their enhanced biocompatibility, especially food-based CDs. Recent studies have shown that food-based CDs can act as carriers for essential nutrient ions (such as Zn), antibiotics medications, therapeutic nucleic acids (e.g., DNA or siRNA), and anticancer drugs [29,84,106,135,136,137,138,139,140,154].

The cellular uptake efficiency of CDs is comparable to that of other nanomaterials and is predominantly influenced by factors such as size, charge, and elasticity [173,174]. Moreover, the nucleus penetration and drug delivery capabilities of GQDs have been attributed to their diminutive size (20–30 nm) and notable elasticity [173]. These accounts underscore the significance of size and elasticity in governing the cellular uptake of nanomaterials. In zebrafish embryos and eleuthero embryos, positively charged CQDs (+45.4 mV; 6.3 nm) with a surface charge of, synthesized using spermidine, exhibited enhanced cellular uptake efficiency and a more pronounced bioaccumulation effect compared to their negatively charged counterparts (−41.3 mV; 4.1 nm) prepared via ammonia citrate. This distinction underscores the facilitating role of positive surface charge in the cellular uptake of CDs [174]. Positively charged nanoparticles are believed to enhance their affinity for negatively charged cell membrane regions, increasing the likelihood of internalization.

Given the remarkable photoluminescent efficacy of CDs, there is a potential to broaden their utility for bioimaging applications [6]. Notably, doping of CDs with heteroatoms like nitrogen (N), phosphorus (P), and sulfur (S) enhances their optical properties [6,67]. This enhancement is attributed to the introduction of defect levels that create additional energy levels and serve as sites for capturing and storing electrons. As a result, during photoexcitation, the facilitated movement and transfer of electrons lead to increased photosensitivity and a higher quantum yield of CDs within the visible light range. For instance, CDs extracted from intricately processed foods, commercially available beverages, fermented culinary items, flavor enhancers, and roasted meats (e.g., duck, chicken, eel, and salmon) not only exhibit consistent and enduring biological imaging performance in in-vitro settings but also hold promise for justification in animal models [30,31,32,33,34,37,38,39,47,48,49,50,51,52,53,54,55,56,57]. The biological imaging application of CDs synthesized from milk, fruits, and edible plants has also been demonstrated in different biological models, including zebrafish, pigs, and mice (Table 2). Likewise, bioimaging application reports extend to CDs synthesized from dietary compounds, such as amino acids, citrate, glutathione, polyamines, and ascorbic acid (commonly known as vitamin C) [122,128,149,160,166,167,168,169]. Among CDs, liposome-like carbon dots known as CD_somes_, synthesized from triolein, demonstrate excitation-dependent fluorescence and exceptional photostability, enabling multi-generation tracking of subcellular organelles for up to six generations after transfer to daughter cells [20]. This demonstrates that the fluorescent properties of CD_somes_ do not interfere with cell division, ensuring biological safety. This highlights the significant potential of CD_somes_ for bioimaging applications. Their remarkable biocompatibility and long-lasting ultra-photostability allow for extended subcellular imaging capabilities without affecting cell division, thereby ensuring biological safety [20].

In the realm of biological imaging, the majority of CDs possess the capability to absorb and emit blue-green light within the ultraviolet spectrum [7,41]. However, this presents a limitation, as the penetration of short-wavelength light in biological tissues is notably inadequate, rendering detection challenging [6]. High-quantum yield CDs offer a partial remedy by enabling in vivo tracking, which mitigates this shortcoming to a certain level [7]. Most of the currently available food-derived CDs belong to this category [41,76]. Furthermore, CDs that emit red and near-infrared (NIR) light are deemed more suitable for applications involving the visualization of biological systems due to their superior tissue penetration capabilities [175]. Presently, no food-based CDs have been reported to exhibit similar imaging capabilities in this context.

### 3.2. Antibacterial Activity

Organic nanomaterials, which encompass lipid- and polymer-nanomaterials, offer inherent antibacterial properties or can serve as carriers for delivering antibacterial agents [4]. In recent years, due to excellent biocompatibility and a simple manufacturing process, CDs have emerged as novel antibiotics [4,10].

Table 4 presents examples of the antibacterial activities of food-based CDs. Studies showed that the synthesized CDs from hydrothermal carbonization of common Chinese herbs, such as Chinese mugwort (*Artemisia argyi*), green chiretta (*Andrographis paniculata*), henna (*Lawsonia inermis*), milk vetch (*Astragalus sinicus*), osmanthus (*Osmanthus fragrans*), rosemary (*Rosmarinus officinalis*), and turmeric (*Curcuma longa*), exhibit antibacterial activity (minimum inhibitory concentrations, MIC_90_ = 0.1–12.0 mg/mL) [88,91,95,98,103,104]. Carbonization products derived from edible parts of plants, such as tea tree (*Camellia sinensis*) leave and garlic (*Allium sativum*) bulbs, have also shown similar results (MIC_90_ = 1.0–10.0 mg/mL) [98,113]. Other investigations reported that certain biological secretions or metabolites with antibacterial activity, including prebiotics secreted by probiotics (e.g., exopolysaccharides) as well as honey produced by bees (e.g., defensin-1), can serve as precursor materials to produce antibacterial CDs [113,114]. Intriguingly, not only food but also antibiotics such as ciprofloxacin and ampicillin have also been employed as raw materials in the synthesis of CDs, leading to antibiotic-functionalized CDs that inherit the functional groups of the antibiotics [159,164]. These antibiotic-functionalized CDs exhibit potent antibacterial activity (0.025–200 μg/mL).

Researchers have shown that the antibacterial activities of CDs can be attributed to their surface charge (Figure 4). Positively charged CDs display significant charge attraction with the negatively charged bacterial cell membrane (with a membrane potential of approximately −100 to −150 mV) [17]. The interaction resulting from this binding can also induce structural damage to the cell membrane [17,127,128,129]. This is in contrast to mammalian cell membranes, which contain higher cholesterol levels and possess a lower membrane potential (around −40 to −80 mV), reducing the likelihood of charge attraction and thus having less impact [17]. An example of this is the investigation conducted by Li et al. [153], who successfully synthesized positively charged CQDs through a two-step carbonization method. Initially, CQDs were formed by pyrolyzing ammonium citrate, which was subsequently mixed with positively charged spermidine, a biological polyamine, and subjected to carbonization. This process resulted in the production of positively charged CQDs with a zeta potential of +60.6 mV [153]. These positively charged CQDs exhibit superior antibacterial activity (MIC_90_ = 0.9 μg/mL) compared to most negatively charged CDs (Table 4). Remarkably, common polyamines such as Put, Spd, and Spm have also been identified as suitable precursors for carbonization, leading to the synthesis of positively charged CQDs. Among them, CQDs-Spd with a zeta-potential of +45.4 mV has demonstrated broad-spectrum antibacterial activity (MIC_90_ = 2.0–4.0 μg/mL) and holds significant therapeutic potential for applications such as bacterial keratitis [124].

Positively charged CDs have also been successfully synthesized using lysine and arginine, which are commonly found in antimicrobial peptides as sources of positively charged amino acids [17,127,128,129]. Studies showed that these CDs inherit positively charged amino acid side chains on the surface and demonstrated remarkable antimicrobial activity with MIC_90_ values ranging from 0.6 to 62.5 μg/mL and exhibited a higher cytotoxicity concentration (CC_50_) along with an excellent selectivity index (MIC_90_/CC_50_). In recent studies, researchers have discovered that CNGs synthesized using lysine exhibit various antibacterial mechanisms [17,127,128,129]. These mechanisms include the positive charge effect on bacterial cell membranes, stimulation of bacterial reactive oxygen species (ROS) production, and attenuation of lysine-based functional groups within the bacterial cell wall structure (Figure 4). Together, these combined effects not only inhibit the production of bacterial drug resistance genes for up to 20 generations but also achieve this inhibitory effect within just 2–3 generations, a significantly faster timeframe compared to antibiotics [127]. The aforementioned advantages highlight the promising potential of antibacterial CDs in the field of biomedicine, generating high anticipation for their future applications.

**Table 4 ijms-24-16579-t004:** Antibacterial activities of food-based CDs.

Precursor	Type/Size (nm)	Zeta-Potential (mV)	Target Bacteria	MIC_90_ /ZOI > 10 mm	Antibacterial Mechanism	Ref.
Chinese mugwort (leaves)	CDs/3.0–7.0	NA	*E. coli* and *S. aureus*	150.0 µg/mL	Inhibition of cell wall synthesis	[78]
Green chiretta (leaf extract)	CDs/8.0–11.0	−3.7	*S. aureus* and *K. pneumonia* (multi-drug resistant clinically isolated strains)	9.6 mg/mL	NA	[81]
Henna (leaves)	CDs/2.7–7.8	−39.0	*E. coli* and *S. aureus*	5.0 mg/mL	NA	[82]
Rosemary (leaves)	CDs/11.5–20.7	NA	*S. aureus*, *B. subtilis*, *Bacillus cereus*, *E. coli*, *S. typhimurium*, and *C. albicans*	12.0 µg/mL	NA	[95]
Tea tree, Osmanthus, or Milk vetch (leaves)	CDs/5.0–18.0	ca. −20	*E. coli* and *S. aureus*	1.0 mg/mL	Cationic effects on bacterial membrane	[98]
Turmeric (rhizome)	CDs/1.5–4.0	−7.5	*E. coli*, *K. pneumoniae*, *S. aureus*, and *S. epidermidis*	250.0–1000.0 µg/mL	ROS generation	[103]
Turmeric (rhizome) + Ammonium persulfate	CDs/9.4–11.8	−17.2	*E. coli* and *L. monocytogenes*	NA	ROS generation	[104]
Oyster mushroom (Sporocarp)	CDs/2.5–5.5	NA	*S. aureus*, *K. pneumoniae*, and *P. aeruginosa*	30.0 µg/mL	ROS generation/Bacterial cell wall damage	[110]
Honey + Garlic	CQDs/4.0–13.0	NA	*E. coli*, *S. aureus*, and *P. aeruginosa*	10.0 µg/mL	Cationic effects on bacterial membrane/ROS generation	[113]
Lysine	CNGs/118.9–178.7	+21.1	*E. coli*, PHBV-producing *E. coli*, CRAB, *S. epidermidis*, *S. aureus*, and MRSA	0.6–10.0 μg/mL	Bacterial cell wall damage/Cationic effects on bacterial membrane/ROS generation	[127]
Lysine or Arginine	CQDs/2.0–7.0	+30.8 or +15.7	*E. coli* and *S. aureus*	16.0–31.3 or 62.5 μg/mL	Cationic effects on bacterial membrane/ROS generation	[129]
Quercetin + Lysine	CNGs/44.8–235.2	+24.2	*E. coli*, *S. enterica*, *P. aeruginosa*, *S. aureus*, and MRSA	0.1–0.9 μg/mL	Bacterial cell wall damage/Cationic effects on bacterial membrane	[17]
Spermidine	CQDs/ ca. 6.0	+45.4	*S. aureus*, MRSA, *E. coli*., *P. aeruginosa*, and *S. Entertidis*	2.0–4.0 µg/mL	Cationic effects on bacterial membrane	[125]
Ammonium citrate/Spermidine	CDs/3.8–5.4	+60.6	*E. coli*, *S. enterica*, *P. aeruginosa*, *S. aureus*, and MRSA	0.9 µg/mL	Cationic effects on bacterial membrane	[153]
Spermine + Dopamine	CQDs/ ca. 10	+31.0	*S. aureus*, MRSA, *E. coli*., *P. aeruginosa*, and *S. entertidis*	2.0–8.0 µg/mL	Cationic effects on the bacterial membrane/Biofilm inhibition	[155]
Ciprofloxacin (antibiotic)	CDs/4.7–6.8	NA	*E. coli* and *S. aureus*	0.025–1.0 µg/mL	NA	[159]
Citric acid + Curcumin	CDs/1.5–2.5	−15.1	*E. coli*, *S. aureus*, *P. aeruginosa*, and *B. subtilis*	375.0–500.0 µg/mL	Cationic effects on bacterial membrane/Biofilm inhibition	[163]
Citric acid + Ethylenediamine/Ampicillin (antibiotic)	CDs/ca. 1.3	−8.0	*E. coli*, *S. aureus*, *P. aeruginosa*, and *B. subtilis*	25.0–200.0 µg/mL	Cationic effects on bacterial membrane/ROS generation	[153]
Citric acid + Branched poly-ethyleneimine	CQDs/2.0–8.0	ca. +15	*S. aureus*	500.0 µg/mL	Cationic effects on bacterial membrane/Biofilm inhibition	[162]
Caffeic acid	CQDs/1.5–2.5	NA	*S. aureus*, *M. luteus*, and *B. cereus*	5.0–10.0 mg/mL	Cationic effects on bacterial membrane	[142]
Fucoidan	CDs/4.0–10.0	−15.8	*E. faecalis*	3.0 mg/mL	ROS generation/Biofilm inhibition	[147]
Citric acid + Urea	CDs/1.0–5.5	−11.6	MRSA and VISA	0.6 µg/mL	NA	[138]
Microcrystalline cellulose	CQDs/5.4–10.2	ca.−10	*E. coli* and *S. aureus*	100.0–350.0 µg/mL	ROS generation/Bacterial cell wall damage	[148]
Triolein	CD_somes_/80.0–100.0	−31.4	*S. aureus*, MRSA, *E. coli*., and *P. aeruginosa*,	1.7–2.5 μg/mL for Gram-positive bacteria; 104.1–112.4 μg/mL for Gram-negative bacteria	Light-triggered ROS generation	[150]
Vitamin C	CDs/3.0–6.0	−20.0	*S. aureus*, *B. subtilis*, *Bacillus* sp. WL-6, and *E. coli*	50.0–75.0 µg/mL	ROS generation	[170]

NA—not available; CDs—carbon dots; CD_somes_—liposome-like carbon dots; CNGs—carbon nanogels; CNPs—carbon nanoparticles; CQDs—carbon quantum dots; CRAB—carbapenem-resistant *A. baumannii*; MRSA—methicillin-resistant *S. aureus;* VISA—vancomycin-intermediate *S. aureus.*

Moreover, the presence of graphene structures within the CD core, either in a sp^2^ or three-dimensional (sp^3^) form, has been demonstrated to utilize mechanical stimulation to act as an electron transfer medium, generating electron-hole pairs [68]. These mobile electrons and holes can interact with nearby oxygen molecules, forming superoxide radicals (^•^O_2_^−^) and hydroxyl radicals (^•^OH), both of which are considered reactive oxygen species (ROS). This phenomenon triggers the production of ROS in bacteria upon contact (Figure 4). An example of this was observed in CDs with intact graphene lattices, obtained through an extended electrolytic reaction involving vitamin C, which exhibits a broad-spectrum antibacterial capability [170]. Another noteworthy example involves CD_somes_, which have been reported to exhibit photocatalytic and peroxidase-imitating activities, which can produce ROS for bacterial eradication driven by a controlled dual-light source [150]. Mechanically, CD_somes_ generate electron-hole pairs under UV irradiation, effectively catalyzing the production of H_2_O_2_. Subsequently, H_2_O_2_ undergoes further decomposition into ^•^OH due to peroxidase-like activity within CD_somes_ when subjected to green light irradiation. Moreover, the bioactivity of triolein-based CD_somes_ in mice models was induced through a sequential dual-light irradiation approach. This includes broad-spectrum antibacterial effects (including against drug-resistant bacteria) and anti-inflammatory properties. Consequently, this leads to accelerated wound healing [150]. However, in the carbonization process, the development of graphene carbon cores in CDs is influenced by several factors, including atom electronegativity, the number of chemical bonds, and the crystal structure of the pristine compounds [67,68]. As a result, the development of novel CDs with fully formed carbon cores encounters difficulties when using certain compounds as pristine compounds. This is particularly true for natural antibacterial compounds like flavones, polyphenols, steroids, and terpenoids. The carbonization of complex compounds can induce cracking reactions, where the rupture of carbon-carbon or carbon-oxygen bonds causes the breakdown of the carbon framework [176]. This breakdown can result in the formation of gaseous products that are released into the reaction, leading to a loss of the carbon source.

Moreover, complex organic compounds may generate unstable intermediates or reaction intermediates, which can engage in alternative chemical reactions, giving rise to undesired side reactions [176]. Therefore, a combination of these natural compounds with substances possessing simpler chemical structures is necessary to facilitate the formation of graphite cores. Examples of such combinations include curcumin/citrate [163], quercetin/lysine [17], and spermine/dopamine [155]. The surfaces of these CDs possess functional groups derived from two distinct sources of raw materials, endowing them with potent antibacterial activity and a wide range of biological functions, including the adsorption of bacterial toxins, antioxidation properties, and anti-inflammatory effects [17,150,155,163]. 

Several studies have shown that food-based CDs can infiltrate biofilms and disrupt bacterial structures. Biofilms consist of a resilient matrix made from extracellular polymeric substances (EPS) self-produced by bacteria. This matrix provides robust protection against environmental threats and serves as an exceptionally effective survival strategy [162]. Additionally, it increases the likelihood of bacteria developing drug-resistant genes [177]. Biofilms create a dense and stable microecological environment, limiting the interaction between bacteria and CDs through material penetration; nonetheless, several food-based CDs have been successfully developed [146,155,162,163,177,178]. Among these CDs, those with a positive charge synthesized from spermine/dopamine and citrate/curcumin have been observed to penetrate the biofilm (typically negatively charged). Subsequently, the cationic effects of these CDs were exerted upon contact with the bacterial cell membrane [155,178]. This process led to the destruction of the bacterial structures and the eventual disintegration of the biofilm (Figure 4). Spermine/dopamine-derived CQDs have also proven effective as lens coatings, effectively inhibiting biofilm formation [155]. This underscores their potential application in preventing biofilms on indwelling medical devices, such as intravascular and urinary catheters. 

Furthermore, within the biofilm, which serves as a hub for bacterial exchange of nutrients, metabolites, and signaling molecules, specific channels that facilitate transport processes are present. These channels are typically characterized by dimensions of approximately 100 nm [162]. These biofilm channels may facilitate the entry of small-sized food-based CDs, such as fucoidan-derived CDs (4.0–10.0 nm in size) and citrate/curcumin-derived CDs (1.5–2.5 nm in size) [130,169], enabling them to exert antimicrobial effects within the biofilm (Figure 4B). The presence of these multifunctional CDs (e.g., antioxidant and anti-inflammatory activity) expands their therapeutic applications in bacterial diseases, such as bacterial keratitis, pneumonia, sepsis, tetanus, and tuberculosis [4].

### 3.3. Antifungal Properties

Fungi are a type of eukaryotic organism known for their chitinous cell wall structure. While certain fungal species, such as Saccharomyces cerevisiae and Aspergillus sojae, have proven beneficial in food processing [179], many others have been identified as harmful to commercial crops, livestock, aquatic organisms, and human health [180]. Examples of these harmful species include *Colletotrichum* spp. (causing anthracnose in plants), *Aphanomyces* spp. (causing red spot disease in fish), Cryptococcus neoformans (causing cryptococcosis in all animals), Candida albicans (causing candidiasis in humans), and Cryptococcus gattii (causing cryptococcal meningitis in humans). To control and combat pathogenic fungi, various small-molecule compounds like amphotericin B, nystatin, and fluconazole have been used [180,181]. However, the administration of these antifungal drugs may carry the risk of nephrotoxicity or potential environmental hazards [180]. 

Numerous natural compounds, including cinnamodial sterols, furanones, eugenol, quinines, and terpenoids, have been extensively studied and identified for their potent antifungal effects [182]. At present, food-derived CDs with comparable antifungal properties remain relatively uncommon, with examples including CDs synthesized from vitamin C or dried fruit powder of Forsythia [86,170]. These food-based CDs are primarily employed in the treatment of plant fungal diseases and are not yet utilized in animals. For instance, CDs derived from vitamin C demonstrate the capacity to infiltrate the nano-sized pores within the fungal cell wall (ca. 10 to 100 nm; material exchange channel) [170]. This enables them to subsequently enter the cell via the endocytosis pathway, where they proceed to exert inhibitory effects on the expression of nonribosomal peptide synthetase genes (synthase for various bioactive molecules, including exopolysaccharides, a major component of the cell wall), consequently influencing fungal growth. Currently, these food-based CDs have not undergone extensive biocompatibility assays and are, therefore, unsuitable for biomedical applications [86,170]. Given the considerable threat posed by various fungal pathogens to human health, it is imperative to explore the development of food-based CDs with efficacy against fungi that infect humans [180]. This could lead to the advancement of novel and potent antifungal treatment strategies in the future.

### 3.4. Antivirus Activity

Table 5 presents the antivirus activities of food-based CDs. In recent studies, a great number of researchers have attempted to explore the antivirus activity of CDs derived from various food additives and dietary compounds, such as boronic acids [156,158], carrageenan [143], glycyrrhizic acid [147], polyethylene glycol (PEG) [165], polyethyleneimine (PEI) [135], pullulan [143], and sodium alginate [157]. It is worth noting that boronic acids, despite being approved by the FDA, have strict limitations regarding their allowable quantity and daily consumption [156,158]. Interestingly, Fahmi et al. [156] employed a two-stage synthesis method to create CQDs with antiviral activity. This process involved the initial carbonization of citric acid into CQDs, followed by the adsorption of boronic acid onto the outer layer. Notably, compared to boronic acid, CQDs exhibited superior antiviral activity (effective concentration, EC_50_ 4.69–9.37 μg/mL) and biocompatibility (CC_50_ > 600 μg/mL). Several dietary supplements, including spermidine, citric acid, folic acid, caffeic acid, lysine, hesperidin, curcumin, and quercetin, have been utilized as raw materials for the development of antiviral CDs (Table 5). These CDs have shown promising antiviral activity against a range of viruses, including the white spot syndrome virus (WSSV), influenza A viruses (IAVs), herpesvirus, rhabdovirus, flavivirus, and coronavirus. To give an example, Lin et al. [157] employed a combination of sodium alginate and ammonium sulfite, followed by carbonization, to facilitate an in situ sulfuration/sulfonation process of the CNGs. In their study, they observed that the carbonization could significantly enhance the biological activities of the CNGs, including their antiviral, antioxidant, and anti-inflammatory properties. Furthermore, the authors observed that the sulfated/sulfonated CNGs demonstrated multifaceted protective effects, safeguarding mice from severe health complications caused by IAVs [157]. Taken together, these studies suggested that the one-step carbonization process and the multiple antiviral mechanisms exhibited by these food-based CDs offer significant advantages for biomedical applications targeting viruses (Table 5).

Extensive research has also been conducted to confirm the presence of specific antiviral-related functional groups in various dietary compounds, including flavonoid compounds (e.g., curcumin, hesperidin, and quercetin) and sulfated polysaccharides [119,120,123,130,157,161]. These functional groups have demonstrated the ability to hinder various stages of viral infection within host cells, including attachment, entry, uncoating, replication, assembly, and release (Figure 5). Viral infection triggers a host defense mechanism, resulting in the production of reactive oxygen species (ROS), intense inflammatory responses, and sustained damage to host cells and tissues [183]. These dietary compounds not only contain antiviral properties but also possess antioxidant and anti-inflammatory activities [176]. Their properties contribute to the reduction of ROS levels and mitigate severe inflammatory responses induced by viruses in host cells [184]. However, the complex chemical structure of these dietary supplements presents a challenge due to their low water solubility and limited bioavailability, which limits their potential for broader biomedical applications [176]. Nevertheless, there is great potential in utilizing a precise and controlled heating process to induce carbonization, thereby modifying their physicochemical properties [119,120,122,130]. This opens up opportunities for improving the bioactive functionalization of CD surfaces. These dietary compounds with multiple biological activities, including antiviral properties, may have some functional groups inherited into food-based CDs through the carbonization process [119,120,123,130,157,161]. In addition, during this process, it is also possible to generate novel functional groups with potent antiviral activity through thermal activation reactions [67,68]. Hence, through conducting numerous experiments under diverse carbonization conditions, it became feasible to identify food-based CDs with enhanced antiviral properties compared to the original dietary compounds [119,120,123,130,157,161]. Moreover, the carbonization process may yield food-based CDs with additional biological activities, such as antioxidants and anti-inflammatory properties [130,157].

**Table 5 ijms-24-16579-t005:** Antivirus activities of food-based CDs.

Precursor	Type/Size (nm)	Target Virus	Toxicity (CC_50_)	Antiviral Effects (EC_50_)	Antiviral Mechanisms	Ref.
Citric acid/Boronic acids	CQDs/5.4–7.0	HIV	>600.0 μg/mL	4.7–9.4 μg/mL	Prevent viral attachment	[156]
Curcumin	CQDs/4.2–5.2	EV71	452.0 μg/mL	0.2 μg/mL	Prevent viral attachment/Inhibition of viral replication	[119]
Curcumin	CQDs/ ca. 4.8	JEV	>100.0 μg/mL	0.9 μg/mL	Prevent viral attachment	[120]
Hesperidin	CPDs/46.7–60.1	EV71	773.0 μg/mL	17.7 μg/mL	Prevent viral attachment/Inhibition of viral replication and translation/Inhibition of viral release/Alleviation of virus-induced oxidation	[123]
Lysine	CNGs/120.0–510.0	IBV (poultry-affecting coronavirus), BEFV (cow-affecting virus), and PRV (pig-affecting virus)	>50.0 μg/mL	<5.0 μg/mL	Prevent viral attachment	[126]
Quercetin	CNGs/326.9–423.3	IAVs	>600.0 μg/mL	0.7 μg/mL	Prevent viral attachment/Alleviation of virus-induced oxidation and inflammation	[130]
Sodium alginate + Ammonium sulfite	CNGs/116.0–183.0	IAVs	>1.0 mg/mL	ca. 250.0 μg/mL	Prevent viral attachment/Inhibition of viral invasion/Alleviation of virus-induced oxidation and inflammation	[157]
Spermidine	CQDs/ ca. 6.0	WSSV (shrimp-affecting virus)	NA	ca. 1.0 μg/mL	Prevent viral attachment/Activation of the immune system	[125]
Vitamin C + PEG-diamine	CDs/4.7	PRRSV (pig-affecting coronavirus)	>250.0 μg/mL	125.0 μg/mL	Inducement of immune defense responses	[165]
Boronic acid derivatives	CQDs/8.9–9.5	HCoV	>100.0 μg/mL	2.0–20.0 μg/mL	Inhibition of the interaction between host cells and viruses/Inhibition of viral replication	[158]
Caffeic acid	CQDs/1.5–2.5	vB-Eos-IME167, T4, and VMY22	>10.0 mg/mL	ca. 2.5 mg/mL	Prevent viral attachment	[142]
Carrageenan or pullulan	CQDs/ ca. 3.1 or ca. 4.2	MERS-CoV	2.0–4.0 μg/mL	ca. 2.5 μg/mL	Prevent viral attachment/Inhibition of viral replication	[143]
Citric acid + Curcumin	CQDs/1.2–1.8	PEDV (pig-affecting virus)	>250.0 μg/mL	ca. 60.0 μg/mL	Prevent viral attachment/Block viral invasion/Inhibition of viral replication/Inhibition of viral release/Alleviation of virus-induced oxidation and inflammation/Stimulation of interferon production	[161]
Glycyrrhizic acid	CQDs/ ca. 11.4	PEDV and PRRSV	>900.0 μg/mL	ca. 300.0 μg/mL	Prevent viral invasion/Inhibition of viral replication/Stimulation of interferon production/Alleviation of virus-induced oxidation	[147]
Citric acid + Urea	GQDs/ 3.0–20.0	feline coronavirus (cat-affecting coronavirus) and EV71	>50.0 μg/mL	ca. 5.0 μg/mL	Prevent viral attachment	[139]
Citric acid + Poly-ethyleneimine	CDs/ca. 3.7	KSHV and EBV	5.0 μg/mL	<5.0 μg/mL	Inhibition of viral replication	[136]

NA—not available; CNGs—carbon nano gels; CPDs—carbonized polymer dots; CQDs—carbon quantum dots; GQDs—graphene quantum dots; BEFV—bovine ephemeral fever virus; E71—Enterovirus 71; EBV—Epstein–Barr virus; HCoV—human coronavirus; HIV—human immunodeficiency virus; IAVs—influenza A virus subtype H1N1; IBV—infectious bronchitis virus; JEV—Japanese encephalitis virus; KSHV—Kaposi’s sarcoma-associated herpesvirus; MERS-CoV—Middle East respiratory syndrome coronavirus; PEDV—porcine epidemic diarrhea virus; PRRSV—porcine reproductive and respiratory syndrome virus; PRV—pseudorabies virus; WSSV—white spot syndrome virus.

### 3.5. Anticancer Activity

CDs derived from food exhibit promising capabilities to facilitate early tumor detection, accurate tumor characterization (e.g., location, size, and type), and enhance tumor treatment (Table 2 and Table 3). These food-based CDs demonstrate potential in various therapeutic aspects, including drug delivery, antiangiogenic effects, promotion of apoptosis, induction of cellular ROS, and boosting immune cell responses (Figure 6). Presently, drug delivery-functional nanomaterials derived from food sources, such as carrot root, citric acid/urea, and citric acid/diethylenetriamine (commonly used chelating agents in food processing), as well as citric acid/tryptophan, uniformly carry a negative charge and effectively employ electrostatic interactions to bind positively charged anticancer drugs (e.g., cisplatin, doxorubicin, and mitomycin) [107,137,142,154]. In addition, food-based CDs with the ability to adsorb negatively charged siRNA have been developed, which require additional modification of PEI on the outer layer [137]. This exemplifies the promising developmental prospects of food-based CDs in the field of gene therapy. Moreover, the majority of food-based CDs employed as drug carriers exhibit the capacity to release cargo (e.g., anticancer drugs) within an acidic environment, aligning with the acidic nature of the microenvironment surrounding cancer cells [137,154]. This property contributes to controlling drug release, thereby enhancing the efficiency of targeted delivery in animal models. In addition, in clinical applications, CDs with a size ranging from 10–200 nm can passively traverse through incomplete vessel walls, which is caused by excessive angiogenesis of cancerous cells and reduce lymphatic drainage within tumor tissue [171,172]. This passive transverse is primarily attributed to the enhanced permeability and retention effect (EPR effect) caused by the excessive angiogenesis around the tumor microenvironment (TME) [185,186]. The EPR effect is a crucial phenomenon in cancer therapy that allows nanomaterials to accumulate in tumors due to their infiltration from disorganized blood vessels [185]. This selective accumulation in tumor tissues enhances the delivery and retention of therapeutic agents, such as anticancer drugs, at the target site while minimizing exposure to healthy tissues [186]. CDs that exploit the EPR effect offer promising opportunities for effective cancer treatment with reduced systemic toxicity [185,186].

Numerous studies have demonstrated that foods contain anticancer properties [183,187]. These properties have also been observed in food-based CDs, such as CDs synthesized from fruits [76], edible plants/fungi [91,96,106,107,110], and spices [118]. Several studies showed that the presence of anticancer properties in these CDs could effectively stimulate the apoptosis of cancer cells (Table 6). Remarkably, the anticancer potential of food-based CDs can be further improved by incorporating them with active substances with known anticancer effects (e.g., microcrystalline cellulose, chitosan, chlorogenic acid, caffeic acid, and quinic acid) as raw materials, which can be attributed to the presence of anticancer-related functional groups and thermal activation reactions that occur during the carbonization process, resulting in novel functional group combinations [52,148,188,189]. This approach enhances the presence of anticancer-related functional groups on the surface of the CDs, thereby increasing their antiangiogenetic properties and immunity-promoting effects [52,67,148,188,189]. For example, Yao et al. utilized three anticancer active compounds in coffee to synthesize CQDs and demonstrated multiple anticancer effects in the hepatoblastoma (HepG2) tumor-bearing mice model, including glutathione (GSH) depletion-dependent ROS production, ferroptosis promotion, and immune cell infiltration [52]. Different from apoptosis, ferroptosis is oxidative stress-dependent programmed cell death, which is regulated by the GSH redox system [189]. Depletion of GSH in programmed cells significantly downregulates glutathione peroxidase, leading to intracellular iron ions triggering the Fenton reaction, which generates lipid peroxides and initiates the production of reactive lipid peroxyl radicals, ultimately culminating in cell death [189]. In a recent study, CD_somes_ synthesized from cooking oil were observed to possess high photocatalytic activity, including photocatalytic oxidase- and peroxidase-like functions when subjected to sequential UV (365 nm) and green light (530 nm) irradiation [151]. Such photo-dynamic reactions prompt cancer cells to generate ROS, leading to the promotion of cancer cell death. This substantiates the potential of CD_somes_ as promising photo-cycling nanozymes suitable for precise tumor phototherapy applications [151]. In summary, exploring the potential of food-based CDs in the above context represents a promising avenue for future development.

**Table 6 ijms-24-16579-t006:** Anticancer activity of food-based CDs.

Precursor	Type/Size (nm)	Cell Strain; Cancer Type	Anticancer Effects (EC_50_)	Anticancer Mechanisms	Ref.
Kiwi, Avocado, or Pear	CDs/4.0–4.5	Caco-2 (colon cancer)/HK-2 (kidney cancer)	**In vitro** 2.2–3.2 mg/mL, 72 h (Caco-2 cells)/1.3–2.0 mg/mL, 72 h (HK-2 cells)	NA	[76]
Green chiretta (leaf extract)	CDs/8.0–11.0	MCF-7 (breast cancer)	**In vitro** 2 mg/mL, 24 h (MCF-7 cells)	NA	[91]
Ginger (rhizome)	CDs/3.5–5.1	A549 (Lung cancer)/MDA-MB-231 (breast cancer)/HeLa (cervical cancer)/HepG2 (liver cancer)/FL83B (liver cancer)	**In vitro** >2.8 mg/mL, 24 h (A549 cells, FL83B cells, and MDA-MB-231 cells)/>0.35 mg/mL, 24 h (HeLa cells)/>1.4 mg/mL, 24 h (HepG2 cells) **In vivo** 440 μg/mice, 16 days, 97% reduction (nude mice; HepG2 cells)	Apoptosis promotion	[101]
Beetroot (root)	CDs/<5.0	MCF-7 (breast cancer)/HepG2 (liver cancer)	**In vitro** 2.7 μg/mL, 24 h (MCF-7 cells) 2.1 μg/mL, 24 h (HepG2 cells)	NA	[106]
Carrot (root)	CDs/ ca. 2.3	MCF-7 (breast cancer)	**In vitro** >1 mg/mL, 24 h (MCF-7 cells)	Anticancer drug delivery (mitomycin)	[107]
Grounded spice of cinnamon, red chili, turmeric or black pepper	CDs/1.0–10.0	LN-229 (brain cancer)	**In vitro** >1–2 mg/mL, 24 h (LN-229 cells; expect for cinnamon CDs)	NA	[118]
Oyster mushroom (Sporocarp)	CDs/5.0–18.0	MDA-MB-231 (breast cancer)	**In vitro** 3.34 μg/mL; 24 h (MDA-MB-231 cells)	Apoptosis promotion	[110]
Carrageenan or pullulan	CQDs/ ca. 3.1 or ca. 4.2	MDA-MB-231 (breast cancer)	**In vitro** ca. 1000 μg/mL; 48 h (MDA-MB-231 cells)	Apoptosis promotion	[143]
Citric acid + Urea	CDs/2.0–6.0	HepG2 (liver cancer)/HeLa (cervical cancer)/MCF-7 (breast cancer)	**In vitro** <2.5 μg/mL (doxorubicin), 48 h (HepG2, HeLa, and MCF-7 cells) **In vivo** 10 mg/mL, 72 h, 50% reduction (HepG2 tumor-bearing mice)	Anticancer drugs delivery (doxorubicin; pH-dependence release)	[140]
Citric acid + Tryptophan	CDs/ ca. 2.6	MGC-083 (gastric cancer)	**In vitro** <1 μM (siRNA), 48 h (MGC-083 cells)	Anticancer drug delivery (siRNA)/Apoptosis promotion	[137]
Citric acid + Diethyl-enetriamine	CDs/5.0–8.0	A2780 and U14 (ovarian cancer)	**In vitro** <11.4 μM (cisplatin), 2 h (A2780 cells) **In vivo** 1.5 mg/mL, 14 days, ~85% reduction (U14 tumor-bearing mice)	Anticancer drugs delivery (cisplatin; pH-dependence release)	[138]
Chlorogenic acid + Caffeic acid + Quinic acid	CQDs/5.0–10.0	HepG2 (liver cancer)	**In vitro** <50 μg/mL, 24 h (HepG2 cells) **In vivo** 25 mg/kg, 12 days, ~80% reduction (HepG2 tumor-bearing mice)	Ferroptosis promotion/ROS induction/Promoting immune cell infiltration	[165]
Glucose + Glutamic acid	CDs/ ca. 2.0	HeLa (cervical cancer)	**In vitro** <0.5 μg/mL (doxorubicin), 48 h (HeLa cells)	Anticancer drugs delivery (doxorubicin; pH-dependence release)	[167]
Microcrystalline cellulose	CQDs/5.4–12.5	HepG2 (liver cancer)	**In vitro** 378.2–482.5 μg/mL, 24 h (HepG2 cells)	Apoptosis promotion/ROS induction	[132]
Triolein	CD_somes_	Tramp-C1 (prostate cancer)	**In vitro** <200 μg/mL, 24 h (Tramp-C1 cells)	ROS induction (photocatalytic activity)	[136]

NA—not available; CDs—carbon dots; CD_somes_—liposome-like carbon dots; CQDs—carbon quantum dots; CNPs—carbon nanoparticles; DOX—doxorubicin.

### 3.6. Immunomodulatory Functions

Table 7 presents the advantages of using food-based CDs in recent years for biomedical applications in the treatment of autoimmune diseases. Researchers have synthesized highly water-soluble CDs from rose petals and employed them as nanocarriers to adsorb thymol, a natural monoterpene that is slightly water-soluble [64]. Thymol with improved bioavailability by CD-based delivery exhibits excellent antioxidant/anti-inflammatory capabilities and effectively alleviates rheumatic symptoms induced by Freund’s complete adjuvant in arthritic rats. Camlik et al. [134] harnessed dietary compound-derived CDs, specifically citric acid and cysteine, to encapsulate insulin and conjugated PEG for the treatment of type I diabetes. This innovative nanostructure effectively shields insulin from the harsh conditions of gastric acid and gastrointestinal enzymes. Remarkably, this approach led to a rapid 60% reduction in blood sugar levels in diabetic mice within 5 h after oral administration [134].

Recent findings indicated that food-based CDs enriched with functional groups derived from edible natural substances hold significant promise in the treatment of challenging autoimmune diseases (Table 7). Traditional Chinese medicines such as *Phellodendri chinensis* cortex and *Rhei radix* rhizoma, which produce CDs through powder carbonization reactions, have shown the capacity to suppress inflammatory factors (e.g., tumor necrosis factor-α, interleukin (IL)-1β, IL-6, matrix metalloproteinase (MMP)-1, MMP-3, and fibroblast growth factor receptor 1) and boost the immunomodulatory functions of anti-inflammatory cytokines (e.g., IL-10 and transforming growth factor-β) [65,66]. These medicines have been applied, for example, in the treatment of autoimmune diseases that affect the skin and gastrointestinal tract, specifically Psoriasis and Ulcerative colitis. In another study, CDs synthesized from folic acid have also been demonstrated to possess anti-inflammatory properties [121]. Folic acid, a member of the vitamin B family, plays a pivotal role in nucleic acid synthesis within the body but lacks anti-inflammatory properties. The anti-inflammatory properties of folic acid-based CDs are attributed to unique functional group combinations that arise as a result of thermal activation during the carbonization process [121]. In an in vivo mouse model of osteoarthritis induced by anterior cruciate ligament transection (ACLT) surgery, it was observed that folic acid-based CDs attenuated the inflammatory response of nuclear factor kappa-light-chain-enhancer of activated B cells (NF-κB)/mitogen-activated protein kinase (MAPK) pathway in chondrocytes when stimulated by the inflammatory factor IL-1β. These CDs also prevented the transition of macrophages into the M_1_ state, which promotes inflammation [121]. Simultaneously, folic acid-based CDs promote the activation of M_0_ phenotype macrophages into the anti-inflammatory M_2_ phenotype, which is capable of releasing anti-inflammatory factors and participating in tissue repair (Figure 7). In addition, food-based CDs show superior clinical potential in combating autoimmune diseases due to their advantages in ease of operation, biocompatibility, and material tunability. These properties make them highly promising for addressing the challenges associated with extremely complex and not fully understood mechanisms of autoimmune disorders.

**Table 7 ijms-24-16579-t007:** Immunomodulatory functions of food-based CDs.

Precursor	Type/Size (nm)	Treatment	Effective Dose	Autoimmune Diseases/Model	Immunomodulatory Mechanism	Ref.
Rose petals/thymol	CDs/5.0–6.0	Oral administration	2 mg/kg	Rheumatoid arthritis/FCA-induced arthritic rats	Drug carrier (thymol)	[64]
*Phellodendri chinensis* cortex	CDs/0.5–3.6	Oral administration	220 μg/kg/day	Psoriasis/IMQ-induced psoriasis-like skin mouse model	Prevent M_1_ transition of macrophages/Activation of M_2_ macrophages	[65]
*Rhei radix* (rhizome)	CDs/1.4–4.5	Oral administration	60 μg/kg/day	Ulcerative colitis/DSS-induced ulcerative colitis mouse model	Inhibition of inflammatory cytokine/Increase antioxidant protein expression level	[66]
Citric acid + Cysteine	CQDs/0.9–1.0	Oral administration	1 mL CQDs solution with 50 IU insulin	Type I diabetes/AOAC standard diet-induced diabetic mice	Drug carrier (insulin)	[134]
Folic acid	CDs/1.0–1.6	Intra-articular injection	2 mg/mL CDs twice per week 6 consecutive weeks.	Osteoarthritis/ACLT mouse model	Inhibition of inflammatory cytokine/Prevent M_1_ transition of macrophages/Activation of M_2_ macrophages	[121]

CDs—carbon dots; CQDs—carbon quantum dots; ACLT—anterior cruciate ligament transection; AOAC—Association of Analytical Communities; DSS—dextran sodium sulfate; FCA—Freund’s complete adjuvant; IMQ—imiquimod.

## 4. Safety Assessment of Food-Based CDs

As food-based CDs hold significant potential for clinical therapeutics, conducting thorough safety evaluations before their widespread adoption becomes crucial [3,4,5]. Furthermore, ensuring the dependable identification of nanomaterial structure and the reproducibility of biological activity is essential for establishing a reliable and comparable foundation, fostering effective communication, and facilitating continuous progress in the field. In pursuit of these goals, Faria et al. [190] proposed that publications in bio-nano research should adhere to the Minimum Information Reporting in Bio-Nano Experimental Literature (MIRIBEL) standard. MIRIBEL guidelines summarize 10 items on material characterization (i.e., composition, size/shape, size dispersity and aggregation, zeta potential, density, drug loading/release, targeting, labeling, and quantification of varied properties), seven items on biological characterization (i.e., cell seeding details, cell characterization, cell/tissue background signal, toxicity studies, justification of biological model, biological fluid characterization, and Animals in Research: Reporting In Vivo Experiments guidelines), and six items on experimental details (i.e., culture dimensions, administered dose, delivered dose, cell/tissue signal with nanomaterial, imaging details, and data statistics and analysis), aiming to have consistent standards from the identification of materials to biological activity and biocompatibility, and even the details of experimental reporting [190].

The toxicity of CDs has been investigated in prior studies, and certain patterns have been identified. In their research, Fan et al. examined 35 distinct CD variations in human macrophages. They discovered that cytotoxicity appears to correlate with smaller size, positive charge, and aggregated tendencies. Conversely, CDs exhibiting a relatively larger size (40–100 nm), neutral charge, and effective dispersion demonstrated excellent biocompatibility [191]. Nevertheless, in vivo, CDs are considered nanomaterials that are readily excreted and metabolized, resulting in low toxicity [192,193,194]. Hence, the in vivo toxicity profiles of CDs exhibit some distinctions. Specifically, CDs with smaller sizes (<6 nm) are observed to prefer renal and hepatic excretion pathways [192]. This preference is attributed to their ability to traverse exclusion barriers within the glomerular filtration assembly and liver sinusoidal endothelial cells (LSECs), facilitating rapid elimination from the body [193,194]. CDs with excessive charges (whether positive or negative) are prone to interact with proteins and other biomolecules, thereby diminishing the excretion rate [192]. 

For current applications in biocompatibility assessment of food-based CDs, a fundamental and straightforward approach to safety assessment involves conducting in vitro cytotoxicity and viability assays. These assays measure cellular metabolic activity or cell proliferation rate to assess potential adverse effects. Another commonly employed assay is the hemolysis assay, which examines the impact of CDs on erythrocytes in biological solutions. However, to comprehensively evaluate the potential hazards of CDs in various application scenarios, in vivo animal models offer a more comprehensive strategy [128]. Animal models enable the pre-evaluation of any possible adverse effects caused by CDs. Researchers have used diverse novel CDs to assess cytotoxicity across different growth stages in various animal models, encompassing embryonic, juvenile, and adult stages (Table 1, Table 2 and Table 3). These comprehensive evaluations contribute to a better understanding of the safety profile of food-based CDs. 

The assessment of embryotoxicity related to the application of food-based CDs has also been investigated. Traditionally, chicken embryos and zebrafish embryos are the most frequently used models for embryonic assessment [82,126,128,166,174]. For example, Chou et al. [126] conducted a study to evaluate the biotoxicity of lysine-carbonized nanogels (Lys-CNGs) using specific-pathogen-free chicken embryos. The authors found that the administered dose of Lys-CNGs did not induce teratogenicity, vascular network formation, or organ structure development in chicken embryos. Regarding zebrafish embryos, researchers have utilized them to evaluate the embryotoxicity of food-based CDs synthesized from various fruits or dietary compounds, including avocado, kiwi, pear, white pitaya, lysine, ammonium citrate, spermidine, and glucose/ethylenediamine [82,126,128,166,174]. The results of the toxicity assessment showed that the half-lethal doses (LC_50_) of these food-based CDs were significantly higher than the effective dose, indicating the high biocompatibility of CDs. Moreover, it was observed that positively charged CDs (i.e., CQDs and CNGs derived from spermidine and lysine) exhibited higher cytotoxicity compared to negatively charged CDs (i.e., CQDs derived from ammonium citrate or glucose/ethylenediamine) under similar experimental conditions [128,166,174]. Furthermore, fluorescence tracking of food-based CDs revealed that positively charged CDs can penetrate the chorion and access the perivitelline space, whereas negatively charged CDs may tend to adhere to the outer surface of the eggshell. This enhanced tissue penetration capability of positively charged CDs in embryos is likely responsible for their increased interference with zebrafish embryonic development [128,174]. Previous studies have also indicated that high concentrations of CNMs, such as GO or single-wall CNTs, can obstruct chorion pore canals, leading to compromised embryonic development due to inadequate nutrient and oxygen supply. This can result in delays or lethality in embryonic development [195,196]. However, when embryos treated with food-based CDs were transferred to a regular culture environment for over 72 h, the fluorescence signal originating from these CDs notably diminished [128,174]. This suggests that rapid clearance may be a prevalent characteristic of CDs, making them more biocompatible compared to other types of nanomaterials.

The eleutheroembryo (larvae) toxicity test has also been done to investigate the toxicity effect of CDs on fish embryos. It is important to note that during the eleuthero embryo stage (72–96 h post-fertilization), the skin and mucus layers of the fish embryo have not yet formed, allowing direct contact with the environment [128,166,174]. This absence of physical barrier protection, like eggshells or mucus, makes the eleuthero embryo highly susceptible to environmental toxins. In recent studies, researchers performed an eleuthero embryo toxicity test to assess the toxicity of food-derived CDs (e.g., CQDs synthesized from avocado, kiwi, pear, or white pitaya,) and dietary compound-derived CDs (e.g., CQDs/CNGs synthesize from lysine, ammonium citrate, spermidine, or glucose/ethylenediamine) on the fish embryo [82,126,128,166,174]. The results indicated that these CDs exhibited lower LC_50_ values in eleuthero embryo toxicity compared to embryotoxicity while remaining well above the effective dose. It is noteworthy that the toxicity studies of these CDs have shown significant variation in the LC_50_ values, which could be attributed to the variations in assay parameters utilized, such as medium composition, embryo stage, treatment duration, and number of embryos used [82,126,128,166,174]. 

Regarding the bioaccumulative distribution of dietary compound-derived CDs, several studies revealed that these food-based CDs were accumulated in various tissues of the eleutheroembryo. These tissues include the eye, lens, tail vessels, yolk sacs, pancreas, and intestines [128,166,174]. However, when these embryos were returned to a normal culture environment for 72 h, the fluorescent signals associated with food-based CNGs in the eleuthero embryo diminished, and the CDs became concentrated solely at the end of the intestine [128,174]. This observation highlights a common trait shared by CDs with distinct structures and raw materials, namely their rapid metabolism and excretion within organisms. Various CNMs, including CDs, CNTs, graphene/GO sheets, and GQDs, have been reported to possess biodegradation capabilities catalyzed by peroxide reductases for degradation [196,197,198,199,200,201]. An notable example in food-based CDs deserves mention. Contemporary degradable CDs with bioactivity are synthesized from vitamin C, which can decompose into CO_2_, CO, and H_2_O within 20 days under room temperature and visible light irradiation [170]. Therefore, we speculate that these food-based CDs may be rapidly eliminated in the zebrafish eleuthero embryo through excretion and peroxide-catalyzed degradation.

The potential toxicity of long-term (>80 days) consumption of dietary compound-derived CDs in the adult stage of zebrafish has also been investigated [128,174]. The investigations showed that the application of CQDs/CNGs synthesized from lysine, ammonium citrate, and spermidine did not cause obvious side effects in adult fish and their offspring. There were no indications of weight loss, reduced mating behavior, decreased egg laying, embryonic growth retardation, or teratogenicity [128,174]. Furthermore, prolonged consumption of Lys-CNGs, which possess broad-spectrum antibacterial properties, did not lead to significant changes in the composition and abundance of intestinal bacteria [128]. Interestingly, researchers observed that Lys-CNGs exhibited bactericidal activity against various bacteria, including multi-drug resistant strains (e.g., methicillin-resistant *S. aureus*). This observation also highlighted the efficient metabolism, excretion, and limited bioaccumulation tendencies of CDs, effectively reducing potential health risks [128]. Certain food-based CDs have also undergone toxicity testing on other aquatic animals. For instance, adult guppy fish (*Poecilia reticulata*) were administered a single super high dose of CQDs obtained from Nescafé^®^ original instant coffee, mixed in equal proportions with commercial feed [39]. These adult fish exhibited survival rates exceeding 4 weeks and no adverse effects.

Biocompatibility evaluation of several food-based CDs has also been conducted using mice models (Table 1, Table 2 and Table 3). In all tested cases, food-based CDs have demonstrated remarkable biocompatibility at effective doses. Regardless of the method of administration, such as oral gavage, intravenous injection, skin sensitization, skin irritation, or spray inhalation, these food-based CDs did not induce any adverse effects on organ structure or biochemical indicators in the mice model (Table 1, Table 2 and Table 3). For example, in one experiment, mice were administered a high dose (2000 mg/kg) of CDs derived from beer or Coke, and the fluorescent signals of CDs were primarily observed in the intestine, liver, and brain within 24 h after CD uptake [37,53]. However, the intensity of the fluorescent signals significantly decreased after 48 h. Importantly, throughout the experimental period, no abnormalities were observed in the major organs or biochemical indicators of the mice [37,53]. It is essential to note that comparing the toxicity variations among different types of CDs on animal models remains challenging due to the lack of standardized testing methods. Considering the extensive application of CDs, particularly in the field of biomedical advancements with their promising prospects [3,4,5,6,7], conducting further safety evaluations to elucidate the potential risks posed by CDs to organisms is warranted. 

In 2017, there was a notable emphasis on the importance of adhering to the Organisation for Economic Cooperation and Development (OECD) guidelines for assessing the toxicity of nanomaterials in animal models [202]. In the OECD guidelines, studies involving nanoparticles in animal models should be followed with the toxicity assessment, including oral toxicity tests (OECD #403, #420, #423, and #425) and sub-chronic toxicity tests (OECD #407 and #408) [128,166,202,203]. Previous studies showed that certain metal nanoparticles, such as silver and copper oxide nanoparticles, caused severe physiological harm to mice, even at low doses ranging from 10–30 mg/kg body weight [203,204]. When it comes to the utilization of food-based CDs, studies have revealed varying toxicity profiles, even for the same CDs, across different animal models. For example, Lin et al. [128] administered an ultra-high dose (2000 mg/kg body weight) of Lys-CNGs in rats for 14 d and performed a toxicity evaluation based on the OECD guidelines #420. The authors observed that these CDs did not cause any physiological harm in their tested adult animals. Toxicity evaluations of Lys-CNGs in other animal models, including zebrafish, guinea pigs, and rabbits, have conformed to OECD specifications for tests and demonstrated excellent biocompatibility, such as the fish embryo acute toxicity test (OECD #236), rabbit skin irritation (OECD #404), and guinea pig skin sensitivity test (OECD #406). Nevertheless, studies have indicated that topical or oral administration of CDs in zebrafish embryos and larvae resulted in adverse effects, whereas such effects were not observed in adult animal models [128,174].

Furthermore, several studies have indicated that enzymes like peroxidase facilitate the excretion and enzymatic breakdown of CDs within the body [128,166,174,196,197,198,199,200,201]. This metabolic process may explain why Lys-CNGs, which exhibit high toxicity in zebrafish embryos, do not induce adverse effects in significantly larger mammals. Although zebrafish models offer advantages such as high throughput and cost-effectiveness, it is crucial to acknowledge that mammalian models, which closely resemble humans, can offer more valuable insights into toxicity evaluation. Therefore, future studies focusing on the utilization of food-based CDs in animal models should incorporate higher-level toxicity assessment methods aligned with international standards, such as OECD guidelines [202]. Adopting these standardized testing methods will empower researchers to make objective comparisons regarding the potential biological toxicity across various CD types, ultimately facilitating a comprehensive understanding of their safety profile.

## 5. Conclusions

This review presents an overview of the emerging applications of food-based CDs and discusses the advantages, challenges, and prospects of these CDs in biomedicine. Our review revealed that food-based CDs, including CQDs, GQDs, CNDs, CPDs, CNGs, CD_somes_, and CNVs, synthesized from various natural resources exhibit unique optical and physicochemical properties. These unique traits have positioned these CDs for a wide array of applications, including antibacterial, antifungal, and antiviral to bioimaging, immunomodulation, and anticancer purposes. Notably, CDs synthesized from pure dietary compounds have shown remarkable therapeutic effects in various disease models.

Furthermore, a comprehensive assessment of food-based CDs in animal models consistently underscores their exceptional biocompatibility and minimal impact on gut flora composition, suggesting their safety for long-term consumption. Rigorous safety evaluations, including testing that adheres to OECD specifications, reinforce the notion that food-based CDs could be regarded as clinically safe drugs with limited environmental impact. To further advance the clinical application of food-based CDs, future studies should explore their biodegradability. Ongoing research and development are needed to investigate the biodegradation mechanisms of various food-based CDs, including self-degradable CDs with multiple bioactivities. These studies will provide valuable insights into potential exposure-related consequences, bolstering confidence in their efficacy and safety profiles. Overall, the findings of this review underscore the promising role of food-based CDs as versatile biomedical prescriptions, opening up exciting possibilities for their utilization in diagnosing and treating various diseases. Further research and development in this field are warranted to fully exploit the potential of food-based CDs and pave the way for their translation into clinical applications.

## Figures and Tables

**Figure 1 ijms-24-16579-f001:**
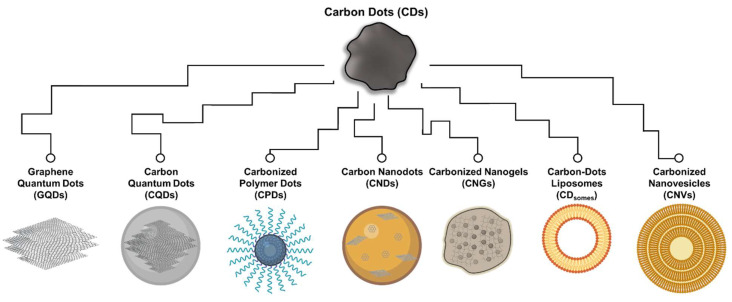
Classification of CDs family. Scheme illustrating the possible structure of various CDs, including graphene quantum dots (GQDs), carbon quantum dots (CQDs), carbon nanodots (CNDs), carbonized polymer dots (CPDs), carbonized nanogels (CNGs), carbon-dot liposomes (CD_somes_), and carbon nanovesicles (CNVs).

**Figure 2 ijms-24-16579-f002:**
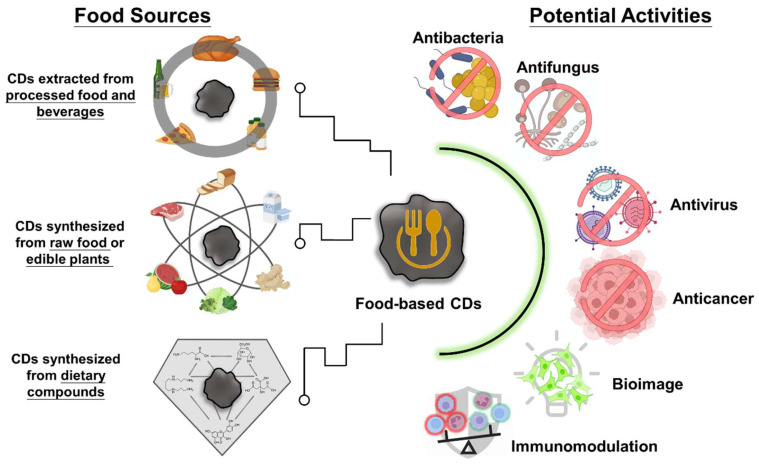
Diagram illustrating the overview of the diverse sources and multiple applications of food-based CDs. Food-based CDs are CDs either extracted from processed food or synthesized using food or dietary compounds in conjunction with food processing-like methods.

**Figure 3 ijms-24-16579-f003:**
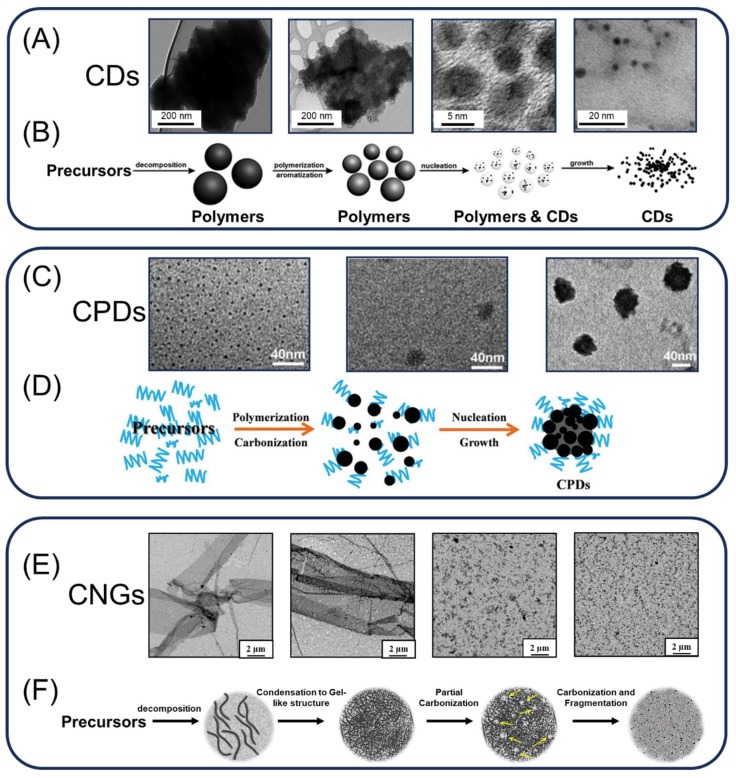
Carbonization step of CDs during the heating process. (**A**) Time-course TEM images of CDs synthesized via hydrothermal carbonization. (**B**) Schematic illustration of the formation mechanism of CDs during carbonization. (**C**) Time-course TEM images of CPDs synthesized from polymer compounds using hydrothermal carbonization. (**D**) The formation mechanism of CPDs from polymer compounds through the carbonization process. (**E**) Time-course TEM images of CNGs synthesized via powder carbonization. (**F**) The carbonization mechanism of the formation of CNGs. (**A**,**B**) reprinted from [44]. (**C**,**D**) reprinted from [10]. (**E**,**F**) reprinted from [17].

**Figure 4 ijms-24-16579-f004:**
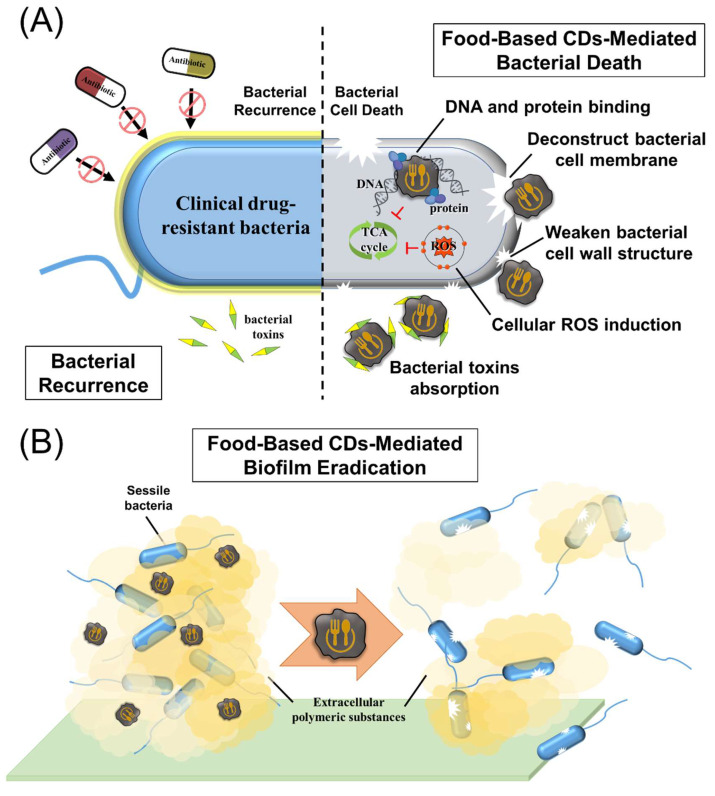
The antibacterial mechanism of food-based CDs. (**A**) Food-based CDs lyse bacteria as a potential pathway to inhibit bacterial damage and drug resistance. The potential harm of drug-resistant bacteria is depicted on the left side of the dotted line. The right side of the dotted line indicates the potential pathways of food-based CD-mediated bacterial death, including DNA and protein binding, deconstructing bacterial cell membrane, weakening bacterial cell wall structure, cellular ROS induction, and bacterial toxins absorption. (**B**) A scheme of food-based CD-mediated biofilm eradication.

**Figure 5 ijms-24-16579-f005:**
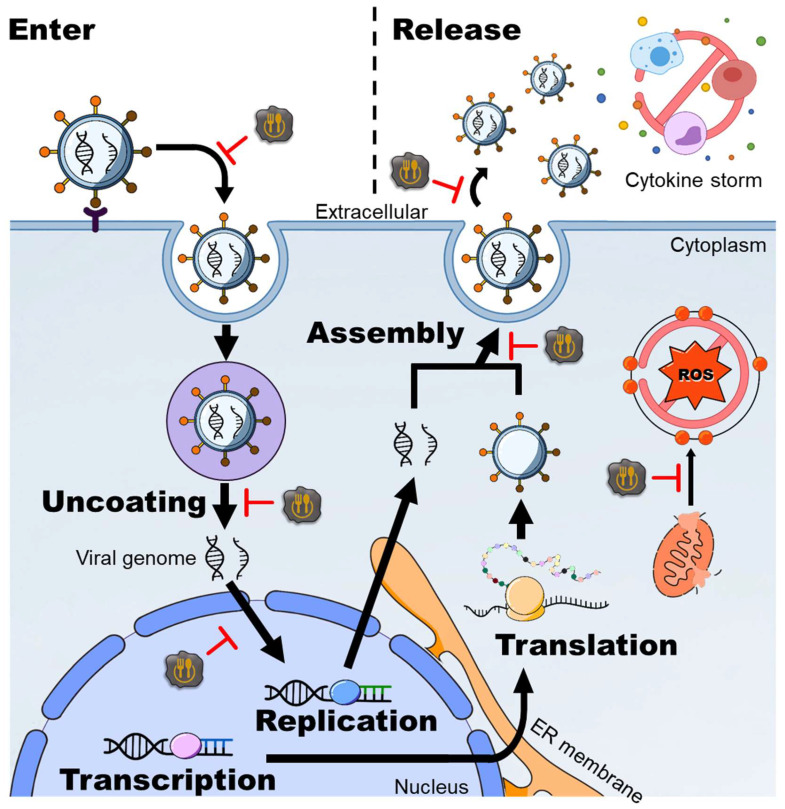
The potential antivirus mechanism via food-based CDs. In addition to limiting the life cycle of viruses, which includes attachment, entry, uncoating, replication, assembly, and release, food-based CDs also curtail the production of intracellular ROS. The inhibition arc is marked in red.

**Figure 6 ijms-24-16579-f006:**
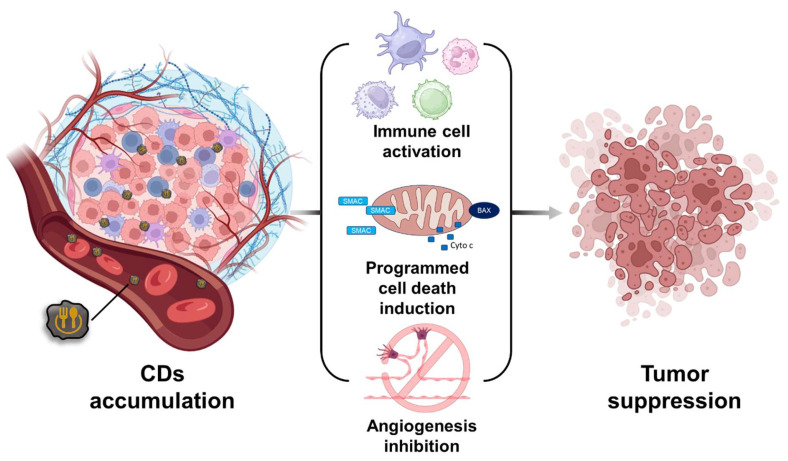
The potential activity of food-based CDs on tumor suppression.

**Figure 7 ijms-24-16579-f007:**
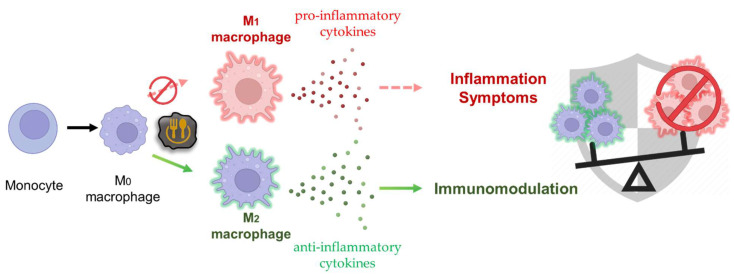
General immunomodulatory activity of food-based CDs. Food-based CDs induce anti-inflammatory mechanisms and promote immunomodulation.

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
