# Peer review of "An Overview of the Potential of Food-Based Carbon Dots for Biomedical Applications"

_ijms, 2023, doi:10.3390/ijms242316579_

Round 1

Reviewer 1 Report

Comments and Suggestions for Authors

In the following submission, An Overview of the Potential of Food-Based Carbon Nano- 2 materials for Biomedical Applications, the authors have comprehensively reviewed the recent progress on food-based carbon nanomaterials (CNMs) and their biomedical applications including antibacterial, antifungal, antivirus, anticancer, and immunomodulation. Although a large number of review articles have been published on carbon-based nanomaterials and their challenges in different applications. But the authors perspective is completely different in this review, as they covered a rarely reported aspect of the carbon materials, i.e., they have covered those carbonaceous materials which have been prepared from food-based substances, which is rarely reported and indeed a need of the hour. This review will surely enhance the knowledge about the carbon materials in the field of biomedical applications The present review article mainly summarizes the recent work and nicely discussed and highlighted the importance of different types of carbon-based materials in aforementioned applications. Thus, it can be accepted for publication in “IJMS”. Some minor comments are as follow

1.          More figures from published studies will enhance the quality of the review

2.          Conclusion should be improved by including future prospects and challenges of carbon materials from food substance and their applications

I.            The authors are suggested to include the following reviews

Lin, Zihong, et al. "Carbon nanomaterial-based biosensors: A review of design and applications." IEEE Nanotechnology Magazine 13.5 (2019): 4-14.

Khan, Mujeeb, et al. "Graphene based metal and metal oxide nanocomposites: synthesis, properties and their applications." Journal of Materials Chemistry A 3.37 (2015): 18753-18808.

Comments on the Quality of English Language

minor corrections

Author Response

Comments from the reviewers and our responses:

We genuinely appreciate the constructive comments and suggestions provided by the reviewers. Response to each comment is given below:

Reviewer #1: In the following submission, “An Overview of the Potential of Food-Based Carbon Nanomaterials for Biomedical Applications”, the authors have comprehensively reviewed the recent progress on food-based carbon nanomaterials (CNMs) and their biomedical applications including antibacterial, antifungal, antivirus, anticancer, and immunomodulation. Although a large number of review articles have been published on carbon-based nanomaterials and their challenges in different applications. But the authors perspective is completely different in this review, as they covered a rarely reported aspect of the carbon materials, i.e., they have covered those carbonaceous materials which have been prepared from food-based substances, which is rarely reported and indeed a need of the hour. This review will surely enhance the knowledge about the carbon materials in the field of biomedical applications The present review article mainly summarizes the recent work and nicely discussed and highlighted the importance of different types of carbon-based materials in aforementioned applications. Thus, it can be accepted for publication in “IJMS”. Some minor comments are as follow

  1. More figures from published studies will enhance the quality of the review

Response: We thank the reviewer for the kind suggestion. We have incorporated a new figure to facilitate the explanation of the CD synthesis process which refer to 3 previous publication.

Figure 3. Carbonization step of CDs during the heating process. (A) Time-course TEM images of CDs synthesized via hydrothermal carbonization. (B) Schematic illustration of the formation mechanism of CDs during carbonization. (C) Time-course TEM images of CPDs synthesized from polymer compounds using hydrothermal carbonization. (D) The formation mechanism of CPDs from polymer compounds through the carbonization process. (E) Time-course TEM images of CNGs synthesized via powder carbonization. (F) The carbonization mechanism of the formation of CNGs. (A) and (B) are refer to [43]. Copyright© 2019, American Chemical Society. (C) and (D) are refer to [10]. Copyright© 2019, Wiley-VCH. (E) and (F) are refer to [17]. Copyright© 2021, Elsevier.

  1. Conclusion should be improved by including future prospects and challenges of carbon materials from food substance and their applications

Response: We appreciate the reviewer's suggestion. We have modified the concolusion as following paragraph. Modified sentences are highlighted in yellow. Please see Line 939-961.

This review presents an overview of the emerging applications of food-derived CDs and discusses the advantages, challenges, and prospects of these CDs in biomedicine. Our review revealed that food-based CDs, including CQDs, GQDs, CNDs, CPDs, CNGs, CDsomes, and CNVs, synthesized from various natural resources exhibit unique optical and physicochemical properties. These unique traits have positioned these CDs for a wide array of applications, including antibacterial, antifungal, and antiviral to bioimaging, immunomodulation, and anticancer purposes. Notably, CDs synthesized from pure dietary compounds have shown remarkable therapeutic effects in various disease models.

Furthermore, a comprehensive assessment of food-based CDs in animal models consistently underscore their exceptional biocompatibility and minimal impact on gut flora composition, suggesting their safety for long-term consumption. Rigorous safety evaluations, including testing that adhere to OECD specifications, reinforce the notion that food-based CDs could be regarded as clinically safe drugs with limited environmental impact. To further advance the clinical application of food-based CDs, future studies should explore their biodegradability. Ongoing research and development are needed to investigate the biodegradation mechanisms of various food-based CDs, including self-degradable CDs with multiple bioactivities. These studies will provide valuable insights into potential exposure-related consequences, bolstering confidence in their efficacy and safety profiles. Overall, the findings of this review underscore the promising role of food-based CDs as versatile biomedical prescriptions, opening up exciting possibilities for their utilization in diagnosing and treating various diseases. Further research and development in this field are warranted to fully exploit the potential of food-based CDs and pave the way for their translation into clinical applications.

  1. The authors are suggested to include the following reviews.

Lin, Zihong, et al. "Carbon nanomaterial-based biosensors: A review of design and applications." IEEE Nanotechnology Magazine 13.5 (2019): 4-14.

Khan, Mujeeb, et al. "Graphene based metal and metal oxide nanocomposites: synthesis, properties and their applications." Journal of Materials Chemistry A 3.37 (2015): 18753-18808.

Response: We sincerely appreciate the reviewer's keen observation of our oversight. We have included the following two citations in the manuscript as Ref 3 and 13.

Reviewer 2 Report

Comments and Suggestions for Authors

1. The novelty and justification of preparing such review is somehow elusive. It is quite obvious that carbon materials can be obtained from most of organic compounds for which carbon atoms are the main constituents (this includes food). Therefore, more justification for such study should be given. Furthermore, I think it would have been more beneficial if the Authors presented the materials grouped by their properties, depending on the carbon source and processing.

2. Overall, the article lacks criticism towards the presented materials and results, doesn’t give much outlook or the Authors personal opinion. Thus, it can hardly be regarded as a review and is rather a compilation of examples. The Authors are kindly referred to read about what a well-constructed review should contain (for example: https://www.ncbi.nlm.nih.gov/pmc/articles/PMC3715443/). Let me just quote: Reviewing the literature is not stamp collecting. A good review does not just summarize the literature, but discusses it critically, identifies methodological problems, and points out research gaps [19]. After having read a review of the literature, a reader should have a rough idea of:

the major achievements in the reviewed field,

the main areas of debate, and

the outstanding research question”

I’m sad to say that upon reading this review I can hardly say to have the rough idea of the above mentioned points.

3. Overall, the study contains very little technical knowledge and shows little criticism to the cited studies. Reading the study I do not get the feeling that the Authors are experts in this field.

4. The Authors treat food-derived CNMs as a unique group of materials with unified properties. Meanwhile, their properties will not depend that much on their source, as on the chemical and structural composition, sizes and shapes, which will vary mostly based on the processing technique. CNMs from different sources which share the same physical properties, will perform in a same matter. In contrast, CNMs obtained from the same source, but processed in a different way, will perform differently. Hence, I find the structure of the study confusing.

5. The study claims to be concerning “carbon nanomaterials”, but in fact, it is mostly about carbon dots or particles. In fact, very few claims that are supposed to be concerning other carbon nanomaterials, such as carbon nanotubes, turn out to be inappropriately referenced. For example, citation 38 does not concern carbon nanotubes but carbon nanomaterials, CNs. Throughout the study, Authors make false or misleading claims, such as that most types of carbon nanomaterials can be synthesized at temperatures close to food-processing temperatures (around 200 0C), which is simply not true for the graphenic materials. In fact, even the cited studies list that temperatures exceeding 500 0C are needed for those. And this is also very optimistic, as in most cases, more than 1000 0C would be needed. It is also not true that pyrolysis typically happens during food processing – pyrolysis requires ambient atmosphere which is not the case for most of the food-processing techniques that are conducted in air.

6. Some criticism toward the cited studies should be employed. If carbon nanoforms are found in so many products, including unprocessed ones, does it mean that they are being synthesized or formed at the place where they’re found, or maybe they are just very common pollutants? To use a simple analogy, microplastic can be found almost everywhere, including woman’s womb. But nobody’s suggesting that it is somehow synthesized within the body.

7. The Authors provide an imprecise and incorrect description of the bottom-up and top-down approaches to synthesize the CNMs. And again, the listed techniques cannot be regarded as universal for all the CNMs types.

8. Throughout the review, no distinction between different carbon nanomaterials’ structures is given. The Authors treat the materials as a group that has unified properties, which is appallingly incorrect and imprecise.

9. The Authors claim that the thermal treatment of food is a green way of synthesizing the carbon nanomaterials. Meanwhile, throughout the study it is proven that it requires: high temperatures, organic solvents, acids, bases, and numerous other synthetic chemicals. How can such energy, substrate, and time- consuming process be regarded as green? This should be elaborated on in the study

10. The Authors over-simplify the CNMs synthesis process by citing some studies that concern carbonization of gels. Different CNMs will require different synthesis procedures – far different from what is claimed in the studies cited from line 86.

11. A lot of studies are improperly referenced, concerning different materials, techniques etc. than what is claimed in the text. In fact, Authors cite a total of 11 studies authored by themselves, and only one can be regarded as concerning food-derived CNMs (citation 105 concerning curcumin, but this is also a big stretch). Rest of the self-citations concern: alginate, quercetin, lysine, polyamines, 1,8-diaminooctane (DAO)/Dextran, and spermidine/ammonium citrate, neither of which is subtracted from food. And in all of these studies, carbon dots or gels are reported with no other forms of nanocarbons studied.

12. Whole section that starts at line 136 contains false claims and incorrect parallels between the CNMs synthesis and chitosan or metallic particles synthesis.

13. At line 164, the Authors claim that to obtain carbon nanomaterials “powders are heated in a muffle furnace at temperatures ranging from 80°C to 350°C for 1 to 8 h [51−53]”. First of all, all of the cited studies concern carbon dots, and not carbon nanomaterials in general. Second of all, no carbon material could be obtained at 80°C, reading through the article, one can easily find that this temperature was used for pre-drying. Lastly, some criticism should be employed while analyzing the studies. In position 51, no direct evidence that the obtained particles are actually carbonaceous is given. In reference 52, CDs are obtained at 400 0C, and the presented results show that they are very rich in oxygen atoms, thus undermining their carbonaceous nature. The last remark is also true for citation 53, which carbonized Rhei radix et rhizome at 350°C.

14. Throughout the study, numerous and unnecessary repetitions can be found. For example, it is quite obvious that all of the CNMs fabrication techniques can be used for each CNMs sources, yet they are listed in every section of the Review. The same is also true for the characterization methods.

15. In some cases, the Authors use oversimplification, inaccurate terms, or state the obvious, all of which should not be present in the scientific literature. For example, carbon materials are not soluble, claim that “CNGs possess a distinct twisted carbon core structure” is inaccurate, or pointing out that carbonization removes some functional groups, retains some, and recombine others is stating the obvious. It is fine to state the obvious only if somebody wants to paint a bigger picture, but just stating these does not show any improvement to the study.

16. The tables are too big and hard to read, I would consider arranging the pages horizontally for these.

17. Some abbreviations are used without being explained, for example, CNGs, HepG2, GSH.

18. At times, it seems like the Authors are assuming that if we take a bioactive compound and carbonize it, we would get a compound with the same activity, but in a form of carbon nanomaterial. This reasoning is unjustified, as in case of biomolecules, bioactivity is governed by a very specific molecular structure. Slight changes to it result in a complete alteration of the overall behavior of the molecule – compare the structure of pseudoephedrine with methamphetamine for a nice example of this.  

19. The study almost completely lacks any explanation of the reported phenomena.

20. Throughout the study, I can see very nice graphics which look like made with some sort of software (looks like Biorender). If that’s the case, usage of the software should be credited in the study.

21. This is probably the first study I have ever seen that suggest that carbonization might make any materials more dispersible in fluids. Typically, it does the opposite and this is the biggest struggle for most of the carbon nanomaterials.

22. The Authors mix intrinsic properties of CNMs with using them as delivery vehicles for  molecules that induce certain effects. These two are not the same nor similar, and listing them together is misleading.

23. In Chapter 4, instead of focusing on the models used and their results, Authors should try to provide and extensive guidelines of the possible determinants of CNMs toxicity.

More detailed remarks can be found as comments to the pdf file attached.

Comments on the Quality of English Language

This study contains some improper usage of words, missing commas, grammar errors, and poorly constructed sentences

Author Response

Comments from the reviewers and our responses:

We genuinely appreciate the constructive comments and suggestions provided by the reviewers. Response to each comment is given below:

Reviewer #2:

  1. The novelty and justification of preparing such review is somehow elusive. It is quite obvious that carbon materials can be obtained from most of organic compounds for which carbon atoms are the main constituents (this includes food). Therefore, more justification for such study should be given. Furthermore, I think it would have been more beneficial if the Authors presented the materials grouped by their properties, depending on the carbon source and processing.

Response: Thanks to the reviewer for the suggestion. CDs synthesized from food materials may offer improved biodegradability and possess cost-effective as well as readily available advantages. In fact, contemporary degradable CDs with bioactivity are synthesized from vitamins, which can decompose into CO2, CO, and H2O within 20 days under room temperature and visible light irradiation [169].

Furthermore, in the revised manuscript, food-based CDs are classified based on heating procedures (i.e., powder carbonization, hydrothermal carbonization, and microwave-assisted hydrothermal carbonization) and carbon sources (processed food, row food, dietary compounds) for their biomedical properties.

Reference

  1. Li, H.; Huang, J.; Song, Y.; Zhang, M.; Wang, H.; Lu, F.; Huang, H.; Liu, Y.; Dai, X.; Gu, Z.; Yang, Z.; Zhou, R.; Kang, Z. Degradable Carbon Dots with Broad-Spectrum Antibacterial Activity. ACS Appl. Mater. Interfaces 2018, 10, 26936–26946, doi:10.1021/acsami.8b08832.

  1. Overall, the article lacks criticism towards the presented materials and results, doesn’t give much outlook or the Authors personal opinion. Thus, it can hardly be regarded as a review and is rather a compilation of examples. The Authors are kindly referred to read about what a well-constructed review should contain (for example: https://www.ncbi.nlm.nih.gov/pmc/articles/PMC3715443/). Let me just quote: “Reviewing the literature is not stamp collecting. A good review does not just summarize the literature, but discusses it critically, identifies methodological problems, and points out research gaps [19]. After having read a review of the literature, a reader should have a rough idea of: the major achievements in the reviewed field, the main areas of debate, and the outstanding research question” I’m sad to say that upon reading this review I can hardly say to have the rough idea of the above mentioned points.

Response: We appreciate the reviewer's suggestion. We have read the articles provided by the reviewers and have substantially revised the manuscript in accordance with the article guidelines. Some critical comments backgroud have been added to the revised manuscript. Please see Line 98-101 and Line 255-269.

  1. Overall, the study contains very little technical knowledge and shows little criticism to the cited studies. Reading the study I do not get the feeling that the Authors are experts in this field.

Response: We value and embrace the reviewer's sincere suggestions. Some technical backgroud have been added to the revised manuscript. Please see Line 239-250, Line 257-260, and Line 284-287.

  1. The Authors treat food-derived CNMs as a unique group of materials with unified properties. Meanwhile, their properties will not depend that much on their source, as on the chemical and structural composition, sizes and shapes, which will vary mostly based on the processing technique. CNMs from different sources which share the same physical properties, will perform in a same matter. In contrast, CNMs obtained from the same source, but processed in a different way, will perform differently. Hence, I find the structure of the study confusing.

Response: We extend our gratitude to the reviewer for highlighting the absence of a definition for CNMs in this manuscript. As per your recommendations, we have made substantial revisions to the manuscript structure. Furthermore, we have also corrected the title as "An Overview of the Potential of Food-Based Carbon Dots for Biomedical Applications."

  1. The study claims to be concerning “carbon nanomaterials”, but in fact, it is mostly about carbon dots or particles. In fact, very few claims that are supposed to be concerning other carbon nanomaterials, such as carbon nanotubes, turn out to be inappropriately referenced. For example, citation 38 does not concern carbon nanotubes but carbon nanomaterials, CNs. Throughout the study, Authors make false or misleading claims, such as that most types of carbon nanomaterials can be synthesized at temperatures close to food-processing temperatures (around 200 ℃), which is simply not true for the graphenic materials. In fact, even the cited studies list that temperatures exceeding 500 ℃ are needed for those. And this is also very optimistic, as in most cases, more than 1000 ℃ would be needed. It is also not true that pyrolysis typically happens during food processing – pyrolysis requires ambient atmosphere which is not the case for most of the food-processing techniques that are conducted in air.

Response: We greatly appreciate the reviewer's confirmation of this well-established scientific fact. In response, we have adjusted the terminology used to describe the food heating process throughout the entire manuscript to "carbonization."

  1. Some criticism toward the cited studies should be employed. If carbon nanoforms are found in so many products, including unprocessed ones, does it mean that they are being synthesized or formed at the place where they’re found, or maybe they are just very common pollutants? To use a simple analogy, microplastic can be found almost everywhere, including woman’s womb. But nobody’s suggesting that it is somehow synthesized within the body.

Response: We value and embrace the reviewer's sincere suggestions. We are grateful for the reviewer's sincere suggestions on the structure of the manuscript. Some critical paragraphs have been added to the revised manuscript. Please see Line Line 98-101 and Line 255-269.

  1. The Authors provide an imprecise and incorrect description of the bottom-up and top-down approaches to synthesize the CNMs. And again, the listed techniques cannot be regarded as universal for all the CNMs types.

Response: We are grateful to the reviewers for pinpointing potential sources of confusion in the manuscript. Indeed, the synthesis method outlined in Lines 109-117 is exclusively suitable for the synthesis of CDs. It's important to note that this manuscript only solely focuses on the synthesis of CDs and does not encompass other types of CNMs. Accordingly, we have adjusted the title, manuscript structure, and replaced instances of ''CNMs'' with ''CDs'' throughout the manuscript. In this modified state, these descriptions of the bottom-up and top-down strategy synthesis make sense.

  1. Throughout the review, no distinction between different carbon nanomaterials’ structures is given. The Authors treat the materials as a group that has unified properties, which is appallingly incorrect and imprecise.

Response: We appreciate the reviewers for the constructive suggestions. In response, we have incorporated the following paragraphs outlining the structural and photoluminescence characteristic differences among various CDs in Line 39-107.

Typical CDs are regarded as organic carbonization products with sizes less than 20 nm, exhibiting excitation-dependent fluorescence properties [7]. They possess sp2/sp3 carbon skeletons and feature an abundance of functional groups and polymer chains within their structures [8]. The surface of CDs is rich in hydrophilic compounds, including carboxyl, hydroxyl, and amine groups, contributing to their excellent water dispersibility [9]. Before 2019, CDs could be further categorized based on their structure (Fig. 1), including graphene quantum dots (GQDs), carbon quantum dots (CQDs), carbon nanodots (CNDs), and carbonized polymer dots (CPDs) [10]. Among these, GQDs are characterized as two-dimensional materials with layered structures typically less than 20 nm in width, generally extending up to 5 layers (ca. 2.5 nm) [11]. Their primary planar structure consists of sp2 carbon hybrid arrangements, predominantly at the edges of graphene sheets or within interlayer defects [12]. Notably, GQDs exhibit distinct graphene lattice structures and a significant presence of chemical groups, particularly oxygen-containing functional groups, contributing to their unique properties, such as the quantum confinement effect and edge effect [13]. CQDs typically assume a spherical shape, with their carbon core primarily featuring excellent sp2 carbon crystallinity and sizes typically falling within the range of 1 to 10 nm [14]. The structural properties of CQDs enable them to exhibit intrinsic state luminescence and size-dependent quantum confinement effect. CNDs closely resemble CQDs in terms of size and shape [15]. They exhibit a higher degree of carbonization, but typically lack a distinct lattice structure and do not manifest the quantum confinement effect related to particle size. The photoluminescence of CNDs stems from defects/surface states and subdomain states within the graphitic carbon core [16]. CPDs are produced through the carbonization of polymer compounds, with a relatively low degree of carbonization in their carbon core [10]. Their primary shared characteristic is the surface of the carbon core being enriched with outward-extending polymer functional groups, a result of passivation during the carbonization process. Regarding photoluminescence of CPDs, mainly originates from surface states, subdomain states, molecular states, and the crosslink-enhanced emission (CEE) effect [10].

Recently, researchers have successfully synthesized and published several new classes of CDs, including carbonized nanogels (CNGs), carbon-dot liposomes (CDsomes), and carbon nanovesicles (CNVs), which share many characteristics in line with CDs but also exhibit distinct structural differences (Fig. 1). CNGs, with sizes ranging from about 100 to 500 nm, are slightly larger than the typically defined in CDs [17,18]. They feature a carbonized structure comprising sp2 conjugated aromatic rings and sp3 polymer structures. Due to their graphene-like embedded polymer structure, CNGs can adopt either spherical or irregular-edged particle forms, displaying physical properties characterized by rheological properties similar to those of flexible polymer structures [17]. The photoluminescent characteristics of CNGs primarily stem from the π-conjugated macrocycle structure and edge chemical functional groups. Specifically, the reduction in vibration and rotation of subfluorophores within crosslinked gel structures is thought to trigger the CEE effect [17,18]. In addition, CDsomes are the carbonization products of long-chain hydrophobic compounds, whose carbon core structure comprises a conjugated benzene ring formed by a blend of sp2/sp3 carbon structure with oxygen-containing functional groups on the surface [19]. A significant characteristic of CDsomes is the asymmetric retention of the aliphatic chain from the precursor on their surface, giving rise to both hydrophilic and hydrophobic properties, rendering them amphipathic CDs with an approximate size of 5 nm. An even more significant and readily observable characteristic is the self-assembly of CDsomes in aqueous solutions, forming vesicles encased by a unilamellar bilayer of amphiphilic CDs. This structure bears a resemblance to liposomes and typically measures around 100 nm in size. The as-formed structure in water is attributed to amphiphilic interactions among the surface ligands, particularly including hydrophobic interactions between the oleate groups [19]. Based on their vesicle structure, CDsomes demonstrate excellent photostability, fusogenicity, and biocompatibility in aqueous solutions. Their excitation-dependent fluorescence properties can be attributed to the presence of polycyclic aromatic clusters, surface emissive traps, and edge defects present in amphipathic CDs of various sizes encapsulated within the vesicle structure [20]. CNVs, as the carbonization products of nonionic surfactants, also possess amphipathic CD characteristics with sizes typically around 3-5 nm [21,22]. Their vesicle structures, however, differ from CDsomes that self-assemble in aqueous solutions, as CNVs feature multilayered bilayer amphipathic CDs and exhibit structural characteristics resembling lipid nanoparticles. Nevertheless, the limited and potent evidence available doesn't allow for a clear differentiation in the mechanistic distinctions of photoluminescence characteristics. Hence, the classification of CDsomes and CNVs remains controversial.

Figure 1. Classification of CDs family. Scheme illustrating the possible structure of various CDs, including graphene quantum dots (GQDs), carbon quantum dots (CQDs), carbon nanodots (CNDs), carbonized polymer dots (CPDs), carbonized nanogels (CNGs), carbon-dot liposomes (CDsomes), and carbon nanovesicles (CNVs).

  1. The Authors claim that the thermal treatment of food is a green way of synthesizing the carbon nanomaterials. Meanwhile, throughout the study it is proven that it requires: high temperatures, organic solvents, acids, bases, and numerous other synthetic chemicals. How can such energy, substrate, and time- consuming process be regarded as green? This should be elaborated on in the study

Response: We agree with the reviewer's feedback and have accordingly eliminated the description of the term "green chemistry" in whole mauescript.

  1. The Authors over-simplify the CNMs synthesis process by citing some studies that concern carbonization of gels. Different CNMs will require different synthesis procedures – far different from what is claimed in the studies cited from Line 86.

Response: We appreciate the insightful suggestions from reviewer, we have reorganized and rephrased the paragraph in Line 86, resulting in the following paragraph, accompanied by relevant reference updates. Please see Line 152-170.

During thermal processing, carbonization reactions are often observed in foods which could result in the formation of CDs [43]. During thermal processing, carbonization reactions are often observed in foods, potentially leading to the formation of CDs [44]. Precursors typically undergo a series of carbonization steps, and certain studies offer time-dependent structural analyses illustrating this evolution (Fig. 3). Initially, the precursor undergoes dehydration, leading to the aggregation and mild condensation of decomposition products, resulting in the formation of large-sized polymer supramolecular structures [44]. With the progression of heating, these polymer supramolecular structures contract due to ongoing intramolecular dehydration. This process is accompanied by the formation of carbon-carbon bonds and the development of aromatic clusters within the polymer. Once the density of clusters reaches a critical supersaturation point, the nucleation of carbon core occurs. At this stage, aromatic clusters diffuse toward the particle surface to form nuclei, and passivation of various functional groups occurs simultaneously [45]. To synthesize CPDs, CNGs, CDsomes, and CNVs, besides selecting suitable precursors, precise control of temperature and heating duration at this stage is crucial [10,17-22]. This control is essential for preserving effective functional groups and stabilizing the carbonized structure. In the case of other types of CDs exhibiting obvious/classical crystal lattices, polymer nanoparticles tend to dissipate or undergo transformation with increasing heating time [44]. This leads to a decrease in the polymer-to-dots ratio, resulting in smaller particle sizes and a narrower distribution of CDs (Fig. 3).

Figure 3. Carbonization step of CDs during the heating process. (A) Time-course TEM images of CDs synthesized via hydrothermal carbonization. (B) Schematic illustration of the formation mechanism of CDs during carbonization. (C) Time-course TEM images of CPDs synthesized from polymer compounds using hydrothermal carbonization. (D) The formation mechanism of CPDs from polymer compounds through the carbonization process. (E) Time-course TEM images of CNGs synthesized via powder carbonization. (F) The carbonization mechanism of the formation of CNGs. (A) and (B) are refer to [43]. Copyright© 2019, American Chemical Society. (C) and (D) are refer to [10]. Copyright© 2019, Wiley-VCH. (E) and (F) are refer to [17]. Copyright© 2021, Elsevier.

  1. A lot of studies are improperly referenced, concerning different materials, techniques etc. than what is claimed in the text. In fact, Authors cite a total of 11 studies authored by themselves, and only one can be regarded as concerning food-derived CNMs (citation 105 concerning curcumin, but this is also a big stretch). Rest of the self-citations concern: alginate, quercetin, lysine, polyamines, 1,8-diaminooctane (DAO)/Dextran, and spermidine/ammonium citrate, neither of which is subtracted from food. And in all of these studies, carbon dots or gels are reported with no other forms of nanocarbons studied.

Response: We sincerely appreciate the thoughtful query from reviewer concerning the definition of food-based CNMs, which aids in avoiding any potential dissemination of unclear or confusing information. We have shifted the title and manuscript structure to focus on the biomedical applications of food-based CDs. Additionally, the generalized foods discussed in the manuscript also contain food additives and functional dietary nutrients, which are regarded as sources of functional groups contributing to the biomedical applications of food-based CDs. Therefore, regarding the dietary compounds or food additives employed in food-based CDs, these include alginate, quercetin, lysine, polyamines, 1,8-diaminooctane/dextran, and spermidine/ammonium citrate, we have the following explanation.

  • Alginate constitutes a significant component in edible seaweeds, accounting for approximately 20% of their dry weight. Furthermore, it serves as a common thickening agent in food processing and can be frequently encountered in products like jelly and ice cream.
  • Quercetin, present in nearly all varieties of fruits and vegetables, is additionally recognized as an edible antioxidant.
  • Lysine, one of the 20 amino acids, is ubiquitous in various foods, encompassing meat, dairy, vegetables, and fruits.
  • Dextran, a complex and branched polysaccharide, is naturally present in various food sources, including refined crystalline sugar, maple syrup, sauerkraut juice, and honey.
  • Ammonium citrate also finds application as a food additive, serving as an acidity regulator, buffer, and emulsifier in products like cheese spreads and chocolate confectionery.
  • Polyamines are naturally occurring compounds present in all living organisms, including foods. They play a multifaceted role in various physiological functions, such as nucleic acids protection, preserving cell membrane integrity, aiding in signal transmission, and facilitating protein transport. Additionally, it's worth noting that spermidine belong to the polyamine family and present almost in all living life, including bacteria, fungi, seaweeds, plnats and animals.

However, 1,8-diaminooctane is a artificially synthetic polyamine and does not in line with the category of natural polyamines (e.g., putrescine, spermidine, and spermine) or common food additives. In response to the reviewer's recommendation, we have consequently eliminated the carbonization products of 1,8-diaminooctane/dextran from the example on food-based CDs. Hence, we regard alginate, quercetin, lysine, polyamines, and spermidine/ammonium citrate as dietary compounds or food additives (Supplementary information, Tables S1 an S2). In agreement with the reviewer's suggestions, we have revised the definition from “food-based CNMs” to “food-based CDs” to establish a more precise classification for these carbonized materials derived from food sources to ensure clarity for readers and avoid any potential misinterpretation.

  1. Whole section that starts at Line 136 contains false claims and incorrect parallels between the CNMs synthesis and chitosan or metallic particles synthesis.

Response: We appreciate the valuable suggestions from the reviewer regarding the manuscript structure. In response, we have removed the mentioned paragraph and revised the subsequent paragraphs as per the recommendations provided in the PDF file. Please see Line 218-230 in our modified manuscript.

The in vivo synthesis of CDs through artificial induction strategies remains an unresolved difficulty. CD synthesis entails the decomposition, dehydration, and polymerization of organic molecules or polymers (Figure 3). Additionally, the verification of the hypothesis regarding CD induction in microorganisms is particularly challenging due to the constraints of in vitro assays in accurately simulating the intricate interplay of multienzymatic dehydration and carbon bond polymerization reactions involved [43,44]. Notably, chitosan can undergo a phyto-synthesis into chitosan particles through a process involving the treatment of plant extracts at 50 °C [60,61]. While chitosan particles lack a crystal lattice and thus cannot be classified as CDs, they still illustrate the potential of bioenzyme catalysis in the production of novel CDs. The reaction is thought to potentially encompass the condensation and polymerization of several enzymes, including nitrate reductase, β-glucosidase, glycolytic enzymes, and aldolases [62]. Consequently, confirming this hypothesis continues to be a significant undertaking in the field of nanomaterial synthesis.

  1. At Line 164, the Authors claim that to obtain carbon nanomaterials “powders are heated in a muffle furnace at temperatures ranging from 80°C to 350°C for 1 to 8 h [51−53]”. First of all, all of the cited studies concern carbon dots, and not carbon nanomaterials in general. Second of all, no carbon material could be obtained at 80°C, reading through the article, one can easily find that this temperature was used for pre-drying. Lastly, some criticism should be employed while analyzing the studies. In position 51, no direct evidence that the obtained particles are actually carbonaceous is given. In reference 52, CDs are obtained at 400 0C, and the presented results show that they are very rich in oxygen atoms, thus undermining their carbonaceous nature. The last remark is also true for citation 53, which carbonized Rhei radix et rhizome at 350°C.

Response: We express our gratitude to the reviewer for their valuable suggestions in this section, encompassing the identification of citation errors, the validation of study accuracy, and constructive comments on the study outcomes. Notably, the appropriate conditions for powder carbonization have been refined to include heating within the range of 180°C to 400°C for a duration of 1-6 h. Furthermore, the subsequent paragraphs have been revised to accommodate constructive critique and feedback on the studies. Please see Line 236-266.

To obtain CDs, from processed edible plants using the dry burning method, these materials are typically dried, and the resulting homogenized powders are heated in a muffle furnace at temperatures ranging from 180°C to 400°C for 1 to 6 h [63−65]. During this heating process, the original material undergoes proper oxidation, facilitating the dehydration and polymerization of various carbon-containing functional groups [23,66]. Applying appropriate heating facilitates the thermal activation of reactions, supplying the necessary energy for chemical interactions between reactant molecules. This promotes the formation of sp2 hybridization and the development of a hexagonal carbon framework [67]. Furthermore, the heating-induced oxidative processes, driven by the electronegativity of oxygen atoms, lead to chemical bonding between carbon atoms, resulting in the introduction of various functional groups and heightened chemical reactivity [68]. However, prolonged heating during dry burning can lead to excessive carbonization, causing the dissipation of non-carbon atoms such as N, O, P, and S [69]. Hence, precise time control during the dry burning process is a critical factor in achieving a rich array of active functional groups on the surface of CDs.

The dry-burning method, also known as powder carbonization, has received limited research attention, and the verification assays related to the graphite lattice are relatively inadequate [63-65]. For example, high-resolution transmission electron microscopy (HR-TEM) analysis of rose CDs fails to discern well-defined graphite lattice patterns (with lattice spacings of 0.246 nm or 0.335 nm), and X-ray diffraction (XRD) data is lacking for the validation of the graphite lattice (with 2θ values of 18.2° or 23.8°) [63]. Additionally, the elemental composition of CDs derived from Rhei radix rhizome and Phellodendri chinensis cortex reveals an excessively high proportion of oxygen elements (24.2% and 28.4%), implying that the synthesis of these CDs may have undergone excessive oxidation [64,65]. Ideal synthesis conditions for these CDs still require further optimization, implying potential progressiveness in the biomedical applications of CDs produced through powder carbonization.

  1. Throughout the study, numerous and unnecessary repetitions can be found. For example, it is quite obvious that all of the CNMs fabrication techniques can be used for each CNMs sources, yet they are listed in every section of the Review. The same is also true for the characterization methods.

Response: We appreciate the reviewers' suggestions on the structure of the manuscript. We have removed redundant sentences throughout the manuscript.

  1. In some cases, the Authors use oversimplification, inaccurate terms, or state the obvious, all of which should not be present in the scientific literature. For example, carbon materials are not soluble, claim that “CNGs possess a distinct twisted carbon core structure” is inaccurate, or pointing out that carbonization removes some functional groups, retains some, and recombine others is stating the obvious. It is fine to state the obvious only if somebody wants to paint a bigger picture, but just stating these does not show any improvement to the study.

Response: We are grateful for the reviewers' suggestions. We have diligently adhered to guidance of reviewer by rephrasing sentences in the PDF file that were identified as unclear.

  1. The tables are too big and hard to read, I would consider arranging the pages horizontally for these.

Response: We have followed the reviewers' suggestions to improve the layout of Tables to improve readability.

  1. Some abbreviations are used without being explained, for example, CNGs, HepG2, GSH.

Response: We appreciate the careful inspection from the reviewers. All text abbreviations have been added. Among them, carbonized nanogels (CNGs) in Line 68; hepatoblastoma (HepG2) tumor in Line 710; glutathione (GSH) in Line 711. We also detected other missing, such as Line 748-752 and Line 760-761.

  1. At times, it seems like the Authors are assuming that if we take a bioactive compound and carbonize it, we would get a compound with the same activity, but in a form of carbon nanomaterial. This reasoning is unjustified, as in case of biomolecules, bioactivity is governed by a very specific molecular structure. Slight changes to it result in a complete alteration of the overall behavior of the molecule – compare the structure of pseudoephedrine with methamphetamine for a nice example of this.

Response: We appreciate the reviewer for highlighting conceptual errors in the manuscript. In fact, the correct concept is as follows, which is Line 334-343 in the revised manuscript.

Several studies have revealed that utilizing dietary compounds in the carbonization reaction enables the inheritance of functional groups onto the surface of the carbon core throughout the heating process, resulting in the formation of novel CDs [117-149]. These functional groups, inherited to the CD surface, may then undergo thermal activation, leading to processes such as residue loss (e.g., decarboxylation and dehydrogenation), rearrangement (e.g., keto-enol tautomerism and conformational changes in aromatic rings), or fusion (e.g., carboxyl fusion and carboxylic acid-amine cross-reaction) [66,67]. The combinations of these newly generated functional groups during the carbonization process could potentially exhibit enhanced biological activities compared to the original bioactive compounds.

  1. The study almost completely lacks any explanation of the reported phenomena.

Response: We are grateful for the reviewer’s suggestion regarding conceptual errors. We have followed the reviewer's instructions in the PDF file to improve the manuscript. In the PDF file, all questions requiring explanation of the mechanism have been resolved. Please also see the response for question 24.

  1. Throughout the study, I can see very nice graphics which look like made with some sort of software (looks like Biorender). If that’s the case, usage of the software should be credited in the study.

Response: We are grateful to the reviewer for noticing our oversight. Illustrations of certain figures was indeed created with BioRender. Hence, we have include the following instructions in the Acknowledgment. Please see Line 971-972.

All types of physiological cells, organisms, food and abstract concept graphics created by BioRender.com (2023). Retrieved from https://app.biorender.com

  1. This is probably the first study I have ever seen that suggest that carbonization might make any materials more dispersible in fluids. Typically, it does the opposite and this is the biggest struggle for most of the carbon nanomaterials.

Response: We appreciate the reviewers' affirmation, which enhances the value of this manuscript.

  1. The Authors mix intrinsic properties of CNMs with using them as delivery vehicles for molecules that induce certain effects. These two are not the same nor similar, and listing them together is misleading.

Response: We have modifed the misleading concept. We have rewritten the relevant sentences. Please see Line 668-664 and Line 698-723.

  1. In Chapter 4, instead of focusing on the models used and their results, Authors should try to provide and extensive guidelines of the possible determinants of CNMs toxicity.

Response: We are grateful for the reviewer’s suggestion regarding conceptual errors. We have rewritten an extensive MIRIBEL guidelines in Chapter 4. Please see Line 781-793 in the modified manuscript.

  1. More detailed remarks can be found as comments to the pdf file attached.

Response: We are grateful to the reviewers for their constructive suggestions on this manuscript. A total of 93 additional suggestions raised by reviewers have been successfully addressed, as recorded below.

  1. Line 18, immunomodulation à anti-immune hyperactivity (Line 18).
  2. Line 19, and Line 19-22, over-simplification à We have modified the sentences in abstract (Line 19-22)
  3. Line 33, In recent years à Nearly 20 years (Line 31).
  4. Line 43-45 and Line 46-47, please see reponse description for question 7.
  5. Line 48-50, we have removed the following statement claiming that the synthesis of CDs is a form of green chemistry. In fact, this false statement has been removed from the entire manuscript.
  6. Line 57-58, temperature/power and time have been attached (Line 123-125).
  7. Line 58-59, change the beginning of the sentence to à Similarities in substrate abundance and synthetic strategies have (Line 125-126)
  8. Line 62, comma missing à roasting, (Line 130)
  9. Line 64, refreshed à regarded (Line 132)
  10. Line 84, pyrolysis à carbonization (Line 151)
  11. Line 86, has been replaced by suitable references (Line 152).
  12. Line 86-92, please see reponse description for question 10 (Line 152-170).
  13. Line 92-94, the misleading sentence has been remove.
  14. Line 96-98, has been unified (Line 183-184).
  15. Line 107, CNMs à CDs (Line 194).
  16. Line 108, the inappropriate reference has been eliminated (Line 194).
  17. Line 110-111, please see reponse description for question 9. Furthermore, organic solvents are appropriate for the extraction and enrichment of CDs already present in food, as they are employed for validation purposes rather than synthesis.
  18. Line 113, typos have been corrected à food-based CDs (Line 200).
  19. Ref 38 in Table 1, CDs type error has been corrected.
  20. Line 134, information has been corrected à …been found to contain CDs, through dialysis and acetonitrile precipitation (Line 217).
  21. Line 136-144 and Line 146, please see reponse description for question 12. We have removed the description of metallic CNMs and rewritten the paragraph (Line 218-230).
  22. Line 147, inappropriate reference has been eliminated.
  23. Line 150-152 and Line 154, a note has been added to the manuscript stating that non-CDs but chitosan particles (Line 225-227).
  24. Line 160, recommendations followed (Line 234).
  25. Line 162, comma missing à CDs, (Line 236)
  26. Line 165, please see reponse description for question 18.
  27. Line 167-169, please see reponse description for question 13. We have rewritten this paragraph to be more precise (Line 234-250).
  28. Line 171, imprecise phrases have been removed.
  29. Line 174, Hydrothermal à Hydrothermal carbonization (Line 267).
  30. Line 176, please see reponse description for question 9. Green chemistry claims have been withdrawn in modified mauescript.
  31. Line 176-177, the nonsensical sentence has been eliminated.
  32. Line 179, changed to "relative high temperatures " to express (Line 271).
  33. Line 189-193, The sentences have been rewritten with more precise wording (Line 281-285).
  34. Line 203, the term natural sweeteners has been used instead (Line 294).
  35. Line 214, Henna is an edible plant that is utilized in traditional Chinese medicine. Different parts of the plant, such as roots, stems, leaves, flowers, and seed powder, are employed in the treatment of various diseases and conditions.
  36. Line 223-226, we have revised the sentence (Line 313-314).
  37. Line 229-231 and Line 231-233, We have removed the imprecise paragraph. In addition, the methods for graphite lattice verification of CDs is mentioned in Line 254-257.
  38. Line 253-255, we have refined the sentences to enhance their rigor and precision (Line 334-343).
  39. Line 256, suggestion has been followed. pyrolytic à carbonization reaction (Line 344).
  40. Line 260, water solubility à water dispersibility (Line 354).
  41. Line 266, Line 271-273, Line 275, Line 277-279, Line 282-284, please see reponse description for question 14. Paragraphs describing repeated concepts have been removed and reorganized (Line 359-363).
  42. Line 296, Activities and potential uses of food-based CNMs à Biomedical applications of food-based CDs (Line 365)
  43. Line 299, we follow the advise from reviewer, adding discussion about dynamin-dependent endocytosis (Line 368-371).
  44. Line 301-302, the sentence has been rewritten to be more precise (Line 374-377).
  45. Line 318-319, since the manuscript has been changed to focus on CDs, the relevant examples of CNTs have been removed.
  46. Line 323-327, size information has been attached (Line 394 and 396).
  47. Line 328, A description of the possible mechanism has been attached (Line 398-400).
  48. Line 330, remove "in the field of biomedicine".
  49. Line 333, the sentence has been rewritten to be more precise on concept description (Line 404-408).
  50. Line 338, we follow the advise from reviewer, applications à justification (Line 411).
  51. Line 341, in vitro examples have been removed (Line 414).
  52. Line 342, activity compounds à dietary compounds (Line 415).
  53. Line 348-350, the sentence has been rewritten to be more precise on mechanical justification (Line 420-425).
  54. Line 351, Line 352-358, Line 358-360, and Line 361-379, this section has been revised to maintain on-topic expression (Line 426-436).
  55. Line 381, the possible mechanisms of CDs have been attatched (Line 459-465, Line 500-504, Line 509-513, and 554-557).
  56. Line 382-383, remove " carbon-based" (Line 438).
  57. Line 392, various higher plants à edible parts of plants (Line 448)
  58. Line 404-406, sentences that lead to misconceptions have been removed.
  59. Line 412-413, the sentence has been rewritten to be more precise on concept description (Line 459-465).
  60. Line 426-428, the sentence has been rewritten and adding the some information to explain the principle (Line 462-468 and Line 484-485).
  61. Line 429, information confirmed.
  62. Line 442-443, the following sentence has been added to provide an improved explanation of the principle (Line 500-502).
  63. Line 449-451, repetitive information has been removed (Line 506-508).
  64. Line 454, peroxidase-mimic metastable states à peroxidase-like active (Line 511).
  65. Line 456, nonsensical sentences have been rewritten (Line 512-513).
  66. Line 459-461, sentence rephrasing complete (Line 516-518).
  67. Line 462-463, sentence rephrasing complete (Line 518-520).
  68. Line 464-466, the sentence has been rewritten to be more precise on concept description (Line 521-527).
  69. Line 474, relevant information is mentioned in the following sentences (Line 535-536)
  70. Line 479-450, sentence rephrasing complete (Line 540-542).
  71. Line 492-493, size informations have been attached (Line 555-556).
  72. Line 494-495, bioacitivty informations have been attached (Line 558).
  73. Line 520-537, section rephrasing complete (Line 582-598)
  74. Line 542, the citation position of the Supplementary Table S1 and S2 has been changed (Line 320).
  75. Line 542-546, irrelevant information has been removed (Line 603-605)
  76. Line 551-555, we apologize for the incomplete sentences and formatting errors here, which have been rewritten (Line 607-612).
  77. Line 563, mid-carbonization à carbonization (Line 620).
  78. Line 567-570, possible mechanisms are explained in the next paragraph (Line 643-662)
  79. Line 582-594, the section describe the antiviral mechanisms of dietary compounds (Line 643-648).
  80. Line 597-600, the sentence has been rewritten and adding the some information to explain the principle Line (648-662).
  81. Line 608-610, please see reponse description for question 18.
  82. Line 617-620, sentence rephrasing complete (Line 672-674).
  83. Line 622, complete information has been attached (Line 679).
  84. Line 624-625, overemphasized sentences has been rewritten (Line 606).
  85. Line 626, supplementary information has been attached (Line 681-682).
  86. Line 641-645, Line 645-648, and Line 649-651, unlike the above food-based CDs mentioned in this section, which do not possess anti-cancer activity; their primary function is drug carriage. Here, we are focusing on food-based CDs with anti-cancer properties. To avoid misunderstandings, we also modify the sentences (Line 703-708).
  87. Line 660, Redundant and repetitive information has been removed (Line 718).
  88. Line 663, to correct information in compliance with reviewer instruction (Line 720-722)
  89. Line 672, we partially agree with the reviewer, but some food-based CDs with immunomodulatory capabilities are also mentioned in this section. Please see Table 7 and Line 669-694 for the 3 examples of food-base CDs with immunomodulatory functions.
  90. Line 700-707, we have added some new sentences describing possible mechanisms (Line 753-758).
  91. Line 717 and Line 718, we have read the reviewer recommended studies and introduced relevant content into the rewritten section (Line 781-796).
  92. Line 726, imprecise wording has been removed.
  93. Line 741-743, imprecise clam has been removed.

Round 2

Reviewer 2 Report

Comments and Suggestions for Authors

The Authors took great care in addressing my comments and I am very satisfied with the way the article reads right now. Some minor issues that still need addressing are given below:

1.       Lines 150 – 153 – there’s a repetition of the same sentence, with different studies cited – please remove the repetition and verify, if the cited studies are correct.

2.       Caption of Figure 3 – the way the proofs are cited should be corrected to be in line with the publisher’s policy. Quoting the exempt from the license to publish the Authors have provided, the following statement should be provided (this applies to all of the reused graphs given in this study – both in the main article and in the supplementary file):

“Reprinted from Publication title, Vol /edition number, Author(s), Title of article / title of chapter, Pages No., Copyright (Year), with permission from Elsevier [OR APPLICABLE SOCIETY COPYRIGHT OWNER]." Also Lancet special credit - "Reprinted from The Lancet, Vol. number, Author(s), Title of article, Pages No., Copyright (Year), with permission from Elsevier.”

3.       As per comment 6 of my previous review, I think it would be beneficial if the Authors mentioned different possibilities as to origins of CDs found in the living organisms (lines 216 – 230). Further, this section contains information about the possibility of CDs being synthesized within the living organism, but the parallel with the chitosan processing is rather confusingly presented. As the Authors say themselves “chitosan can undergo a phyto-synthesis into chitosan particles through a process involving the treatment of plant extracts at 50 °C”. This is hardly an in vivo condition. In fact, both of the cited studies use plant extracts as “templates” for precipitating chitosan from its acid solution. Hence, I think that this section requires some further clarification as to avoid confusion.  

4.       As I previously stated, when discussing the possible toxicity of the CDs, apart from analyzing how it is studied, a general consensus as to what are the toxicity determinants, and what properties should CDs have that could promise their safe application should be given. There must already be some consensus on that matter, regarding the size, shape, functional groups etc. I believe that such information would strengthen the study and provide the readership with important guidelines as to what to aim for.

Comments on the Quality of English Language

In line 971-972, a verb is missing from the passive sentence “All types of physiological cells, organisms, food and abstract concept graphics created by BioRender.com (2023).”

Author Response

2nd round for Reviewer #2:

  1. Lines 150 – 153 – there’s a repetition of the same sentence, with different studies cited – please remove the repetition and verify, if the cited studies are correct.

Response: We appreciate that the reviewer noticed our oversight. We have removed repetitive sentences and cited the correct reference. Please see Line 150-152.

  1. Caption of Figure 3 – the way the proofs are cited should be corrected to be in line with the publisher’s policy. Quoting the exempt from the license to publish the Authors have provided, the following statement should be provided (this applies to all of the reused graphs given in this study – both in the main article and in the supplementary file):

 “Reprinted from Publication title, Vol /edition number, Author(s), Title of article / title of chapter, Pages No., Copyright (Year), with permission from Elsevier [OR APPLICABLE SOCIETY COPYRIGHT OWNER]." Also Lancet special credit - "Reprinted from The Lancet, Vol. number, Author(s), Title of article, Pages No., Copyright (Year), with permission from Elsevier.”

 Response: We have followed the reviewer’s instructions. The following paragraph have been added to the end of Fig. 3 Caption (Line 178-186).

(A)  and (B) reprinted from ACS Omega, 4, Papaioannou, N.; Titirici, M.-M.; Mahmoudi, E.; Sapelkin, A. Investigating the Effect of Reaction Time on Carbon Dot Formation, Structure, and Optical Properties. 21658–21665, Copyright© 2019, with permission from American Chemical Society. (C) and (D) reprinted from Advanced Science, 6, Xia, C.; Zhu, S.; Feng, T.; Yang, M.; Yang, B. Evolution and Synthesis of Carbon Dots: From Carbon Dots to Carbonized Polymer Dots. 1901316, Copyright© 2019, with permission from Wiley-VCH. (E) and (F) reprinted from Chemical Engineering Journal, 411, Lin, H.-Y.; Wang, S.-W.; Mao, J.-Y.; Chang, H.-T.; Harroun, S.G.; Lin, H.-J.; Huang, C.-C.; Lai, J.-Y. Carbonized Nanogels for Simultaneous Antibacterial and Antioxidant Treatment of Bacterial Keratitis. 128469, Copyright© 2021, with permission from Elsevier.

  1. As per comment 6 of my previous review, I think it would be beneficial if the Authors mentioned different possibilities as to origins of CDs found in the living organisms (lines 216 – 230). Further, this section contains information about the possibility of CDs being synthesized within the living organism, but the parallel with the chitosan processing is rather confusingly presented. As the Authors say themselves “chitosan can undergo a phyto-synthesis into chitosan particles through a process involving the treatment of plant extracts at 50 °C”. This is hardly an in vivo condition. In fact, both of the cited studies use plant extracts as “templates” for precipitating chitosan from its acid solution. Hence, I think that this section requires some further clarification as to avoid confusion.  

 Response: We appreciate the reviewer for bringing attention to this potentially confusing paragraph. Unfortunately, as of now, no studies have discovered CDs produced by living organisms themselves. Consequently, it is currently not feasible to add a new reference at this stage. Regarding the latter question, we have identified the problematic sentence and refined the information for clarity. Please see Line 229-233.

The in vivo synthesis of CDs through artificial induction strategies remains an un-resolved difficulty. CD synthesis entails the decomposition, dehydration, and polymerization of organic molecules or polymers (Figure 3). Additionally, the verification of the hypothesis regarding CD induction in microorganisms is particularly challenging due to the constraints of in vitro assays in accurately simulating the intricate interplay of multienzymatic dehydration and carbon bond polymerization reactions involved [43,44]. As of now, no studies have discovered CDs produced by living organisms themselves. Notably, through a phyto-synthesis process at 50 °C using plant leaf extracts in an in-vitro environment with chitosan dissolved in an acidic solution, chitosan particles (ca. 10 nm) are synthesized. [60,61]. While chitosan particles lack a crystal lattice and thus cannot be classified as CDs, they still illustrate the potential of bioenzyme catalysis in the production of novel CDs. The reaction is thought to potentially encompass the condensation and polymerization of several enzymes, including nitrate reductase, β-glucosidase, glycolytic enzymes, and aldolases [62]. Consequently, confirming this hypothesis continues to be a significant undertaking in the field of CD synthesis.

  1. As I previously stated, when discussing the possible toxicity of the CDs, apart from analyzing how it is studied, a general consensus as to what are the toxicity determinants, and what properties should CDs have that could promise their safe application should be given. There must already be some consensus on that matter, regarding the size, shape, functional groups etc. I believe that such information would strengthen the study and provide the readership with important guidelines as to what to aim for.

 Response: Thanks to the reviewer for the suggestion. We can add the following paragraph with newly cited references to the manuscript. Please see Line 804-817.

The toxicity of CDs has been investigated in prior studies, and certain patterns have been identified. In their research, Fan et al. examined 35 distinct CD variations in human macrophages. They discovered that cytotoxicity appears to correlate with smaller size, positive charge, and aggregated tendencies. Conversely, CDs exhibiting relative larger size (40-100 nm), neutral charge, and effective dispersion demonstrated excellent biocompatibility [191]. Nevertheless, in vivo, CDs are considered nanomaterials that are readily excreted and metabolized, resulting in low toxicity [192-194]. Hence, the in-vivo toxicity profiles of CDs exhibit some distinctions. Specifically, CDs with smaller sizes (< 6 nm) are observed to prefer renal and hepatic excretion pathways [192]. This preference is attributed to their ability to traverse exclusion barriers within the glomerular filtration assembly and liver sinusoidal endothelial cells (LSECs), facilitating rapid elimination from the body [193,194]. CDs with excessive charges (whether positive or negative) are prone to interact with proteins and other biomolecules, thereby diminishing the excretion rate [192].

Reference

  1. Fan, J., Claudel, M., Ronzani, C., Arezki, Y., Lebeau, L., Pons, F. Physicochemical Characteristics That Affect Carbon Dot Safety: Lessons from A Comprehensive Study on A Nanoparticle Library. Int. J. Pharm. 2019, 569, 118521, doi:10.1016/j.ijpharm.2019.118521.
  2. Truskewycz, A., Yin, H., Halberg, N., Lai, D.T.H., Ball, A.S., Truong, V.K., Rybicka, A.M., Cole, I. Carbon Dot Therapeutic Platforms: Administration, Distribution, Metabolism, Excretion, Toxicity, and Therapeutic Potential. Small 2022, 18, 2106342, doi:10.1002/smll.202106342.
  3. Liang, X., Wang, H., Zhu, Y., Zhang, R., Cogger, V.C., Liu, X., Xu, Z.-P., Grice, J.E., Roberts, M.S. Short- and Long-Term Tracking of Anionic Ultrasmall Nanoparticles in Kidney. ACS Nano 2016, 10, 1, 387–395, doi:10.1021/acsnano.5b05066.
  4. Tsoi, K.M., MacParland, S.A., Ma, X.-Z., Spetzler, V.N., Echeverri, J., Ouyang, B., Fadel, S.M., Sykes, E.A., Goldaracena, N., Kaths, J.M., Conneely, J.B., Alman, B.A., Selzner, M., Ostrowski, M.A., Adeyi, O.A., Zilman, A., McGilvray, I.D., Chan, W.C.W. Mechanism of Hard-Nanomaterial Clearance by the Liver. Nat. Mater. 2016, 15, 1212–1221, doi:10.1038/nmat4718.
